# Generative Models are Self-Watermarked: Declaring Model Authentication through Re-Generation

**Aditya Desu**[*]                                                        *adityadesu100@gmail.com*
*The University of Melbourne*

**Xuanli He**[*]                                                          *z.xuanli.he@gmail.com*
*University College London*

**Qiongkai Xu**[†]                                                        *qiongkai.xu@mq.edu.au*
*Macquarie University*
*The University of Melbourne*

**Wei Lu**                                                               *luwei@sutd.edu.sg*
*Singapore University of Technology and Design*
*Skywork AI*

**Reviewed on OpenReview:** *https://openreview.net/forum?id=LUHmWDydue*

## Abstract

As machine- and AI-generated content proliferates, protecting the intellectual property of generative models has become imperative, yet verifying data ownership poses formidable challenges, particularly in cases of unauthorized reuse of generated data. Confirming the ownership of the data is challenging, as the data generation process is opaque to those verifying the authenticity. Our work is dedicated to detecting data reuse from a single sample. While watermarking has been the traditional method to detect AI-generated content by embedding specific information within models or their outputs, which could compromise the quality of outputs, our approach instead identifies inherent fingerprints in the outputs without altering models. The verification is achieved by requiring the (authentic) models to re-generate the data. Furthermore, we propose a method that iteratively re-generates the data to enhance these fingerprints in the generation stage. The strategy is both theoretically sound and empirically proven effective with recent advanced text and image generative models. Our approach is significant because it avoids extra operations or measures, such as (1) modifying model parameters, (2) altering the generated outputs, or (3) employing additional classification models for verification. This enhancement broadens the applicability of authorship verification (1) to track the IP violation in generative models published without explicitly designed watermark mechanisms and (2) to produce outputs without compromising their quality.

## 1 Introduction

In recent years, the emergence of Artificial Intelligence Generated Content (AIGC), including tools like ChatGPT, Claude, DALL-E, Stable Diffusion, Copilot, has marked a significant advancement in the quality of machine-generated content. While these cloud-based services have accelerated AI technology development, they have also raised concerns about content misuse. Two critical challenges emerge: (1) protecting the Intellectual Property (IP) of authentic generators and (2) ensuring accountability for information sources.

---

[*]Equal contribution.
[†]The corresponding author.

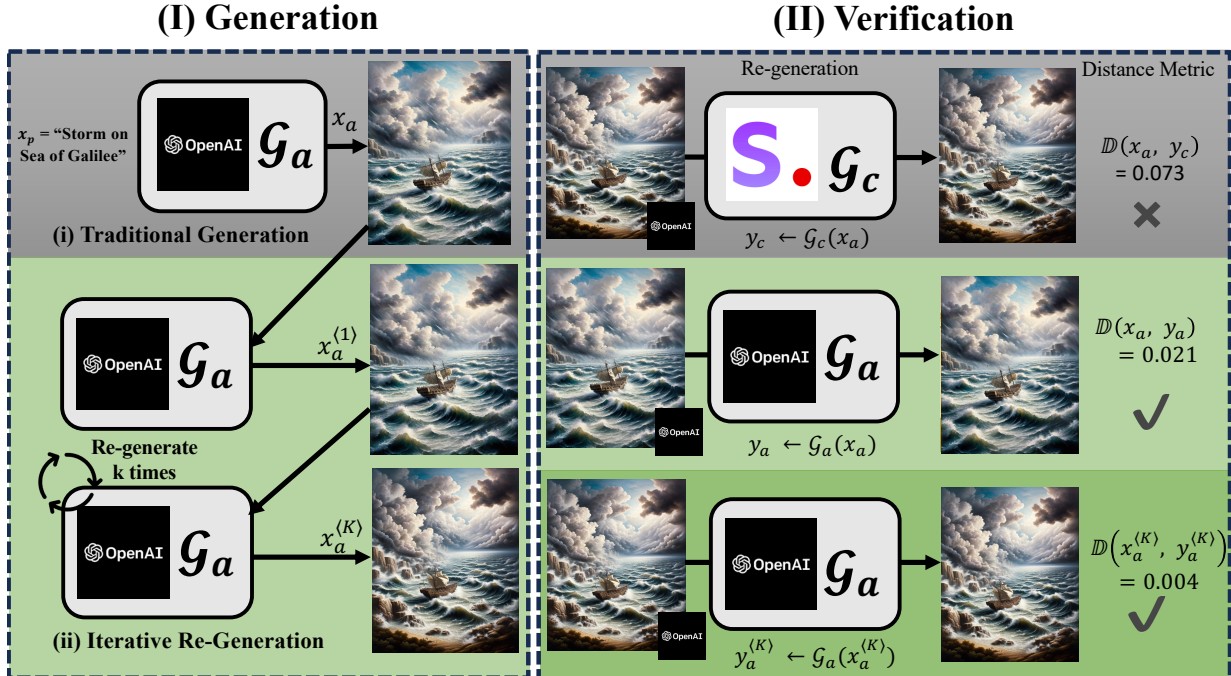

Figure 1: The two-stage framework leveraging fingerprints in generative models. In *(I) **Generation Stage***, models generate output in traditional ways and optionally re-generate the output $k \in [1..K]$ times to enhance the fingerprints. The process ensures consistency of the modality in iterative re-generation, *i.e.,* text-to-text and image-to-image. In *(II) **Verification Stage***, the authentication of data ownership is established by assessing the distance between the suspected data (left) and its re-generated version (right). There exists a distinguishable margin between the distances by authentic generator ($\mathcal{G}_a$, the middle and the bottom examples) and contrasting benign generator ($\mathcal{G}_c$, the top example), exemplified by models from OpenAI ($\mathcal{G}_a$) and Stability AI ($\mathcal{G}_c$), respectively.

Traditionally, the primary approach for safeguarding IP of the contents generated by AI generator has involved embedding subtle but verifiable watermarks into their outputs, such as text (He et al., 2022a;b; Kirchenbauer et al., 2023a), images (Zear et al., 2018; Zhao et al., 2023b) and code (Lee et al., 2023). These watermarking techniques typically involve adding supplementary information to the deep learning model's parameters and architectures or direct post-processing alterations to the generated outputs. However, these alterations could potentially degrade the quality of the generated content. An alternative strategy has been the classification of data produced by a specific model to distinguish it from content generated by other models or humans (Solaiman et al., 2019b; Ippolito et al., 2020). Nonetheless, this often requires training additional classifiers to verify authorship, raising concern about their ability to generalize and maintain robustness across evolving generative models, especially with few training samples.

Another approach has been to classify data from a specific model to tell it apart from data created by other models or humans (Solaiman et al., 2019b; Ippolito et al., 2020). However, this usually needs extra classifiers to confirm authorship, which can be challenging. These classifiers might struggle to stay effective and robust as generative models evolve, especially with limited training samples.

In response to the challenges of authorship authentication and IP protection, our approach is to exploit the inherent characteristics of generative models. Firstly, we recognize that generative models possess unique attributes - akin to model fingerprints - such as specific styles and embedded knowledge. In the *Verification Stage* of our framework, we utilize these implicit fingerprints by measuring the *distance* $\mathbb{D}$ between the genuine data samples with content re-generated by the authentic and contrasting models. Secondly, to enhance the distinctive nature of these fingerprints, our approach in the *Generation Stage* involves using the original

model to iteratively re-generate outputs from previous iterations. This process is grounded in fixed-point theory (Granas & Dugundji, 2003) and its practical applications. Through this iterative re-generation, the model's inherent fingerprints become more evident, enhancing the effectiveness of our verification process.

In Figure 1, we present a conceptual framework for authorship verification through re-generation.

**Stage I: Generation**  The authentic generator targets producing outputs that involve stealthy but significant signatures that are distinguishable from other generative models or humans. We consider two distinct approaches: *(i) 'Traditional Generation'* produces the authentic outputs $\boldsymbol{x}_a$ from a given text input as a prompt, *i.e.,* $\boldsymbol{x}_a = \mathcal{G}_a(\boldsymbol{x}_p)$, where $\boldsymbol{x}_p$ = "Storm on Sea of Galilee"; and *(ii) 'Iterative Re-Generation'* enhances the model's unique signature by re-generating the data multiple times using a 're-painting' or 'paraphrasing' mode, *i.e.,* $\boldsymbol{x}_a^{\langle k+1 \rangle} = \mathcal{G}_a(\boldsymbol{x}_a^{\langle k \rangle})$. Here $\mathcal{G}_a$ is the authentic generative model which is DALL·E (Ramesh et al., 2021) in this example.

**Stage II: Verification**  In this stage, authentic model $\mathcal{G}_a$ verifies the origin of its artefact $\boldsymbol{x}_a$ by comparing the 'distance' $\mathbb{D}$ between $\boldsymbol{x}_a$ and its re-generation by the authentic model $\mathcal{G}_a(\boldsymbol{x}_a)$ or other contrasting models $\mathcal{G}_c(\boldsymbol{x}_a)$. Intuitively, the one-step regeneration distance of an image originally generated by the authentic model, such as DALL·E by OpenAI, is expected to be smaller when compared to itself than to a contrasting model not involved in its initial generation, *i.e.,* $\mathbb{D}(\boldsymbol{x}_a, \mathcal{G}_a(\boldsymbol{x}_a)) < \mathbb{D}(\boldsymbol{x}_a, \mathcal{G}_c(\boldsymbol{x}_a))$. Furthermore, the more re-generations an image undergoes during the *Generation Stage*, the lower its one-step regeneration distance of the authentic model becomes at the *Verification Stage*, *i.e.,* $\mathbb{D}(\boldsymbol{x}_a^{\langle i \rangle}, \mathcal{G}_a(\boldsymbol{x}_a^{\langle i \rangle})) < \mathbb{D}(\boldsymbol{x}_a^{\langle j \rangle}, \mathcal{G}_a(\boldsymbol{x}_a^{\langle j \rangle}))$, when $i > j$.

We summarize the key advantages and contributions of our work as follows:

- We validate the effectiveness of using re-generated data as a key indicator for authorship verification. This approach is designed to be functional in black-box settings and applies across various generative applications, such as Natural Language Generation (NLG) and Image Generation (IG).

- We introduce an iterative re-generation technique to enhance the inherent fingerprints of generative models. We use fixed-point theory to demonstrate that modifications achieved through re-generation converge to minimal edit distances. This ensures a distinct separation between the outputs of authentic models and those generated by other models.

- We have developed a practical verification protocol that streamlines the process of data ownership validation in generative models. This protocol is especially useful in legal settings, as it eliminates the need for generators to disclose their model parameters or watermarking strategies, thereby preserving confidentiality and proprietary integrity.

- A notable advantage of our approach is its reliance solely on the standard generative models, without resorting to additional interventions, including (1) manipulating or fine-tuning generative model parameters, (2) post-editing the outputs, or (3) additional independent classification models for verification. This simplicity in design not only preserves the original quality of the generated content but also enhances the feasibility and accessibility of our verification method.

## 2   Related Work

Recent advancements in the field of generative modeling, exemplified by innovations like DALL · E (Ramesh et al.), Stable Diffusion (Rombach et al., 2022), ChatGPT, Claude, Gemini, and others. However, this proliferation of synthetic media has simultaneously raised ethical concerns. These concerns include the potential for misuse in impersonation (Hernandez, 2023; Verma, 2023), dissemination of misinformation (Pantserev, 2020; Hazell, 2023; Mozes et al., 2023; Sjouwerman, 2023), academic dishonesty (Lund et al., 2023), and copyright infringement (Brundage et al., 2018; Rostamzadeh et al., 2021; He et al., 2022a; Xu & He, 2023). In response, there is an increasing focus on the need to trace and authenticate the origins of such content to prevent its illegitimate use in AIGC. Considering the distinct nature of image and text, we will review the authorship identification in image and text generation models separately.

**Authorship Identification for Image Generation Models**

Traditional image watermarking methods involve imprinting unique watermarks onto generated images, either through direct alterations to pixel values in the spatial domain or by embedding watermarks into altered image forms in the frequency domain (Cox et al., 2008). Recent deep learning techniques have proposed leveraging neural networks to encode concealed information within images (Zhu et al., 2018; Yang et al., 2019; Ahmadi et al., 2020; You et al., 2020). These methods ensure that subsequent transformations applied to watermarked images preserve the integrity of the embedded information (Fernandez et al., 2022; 2023). However, given the escalating concerns regarding deep fakes (Brundage et al., 2018; Harris, 2019), new methodologies have been proposed to attribute the origin of an image by identifying subtle, machine-detectable patterns unique to GAN-generated images (Marra et al., 2019; Afchar et al., 2018; Güera & Delp, 2018; Yu et al., 2019; Zhang et al., 2019; Durall et al., 2020; Liu et al., 2020).

Similarly, Fernandez et al. (2023) introduce a binary signature directly into the decoder of a diffusion model, resulting in images that contain an imperceptibly embedded binary signature. This binary signature can be accurately extracted using a pre-trained watermark extractor during verification. Given the escalating concerns regarding the misuse of deep fakes, as highlighted in the literature (Brundage et al., 2018; Harris, 2019), several studies have proposed methodologies for attributing the origin of an image, specifically discerning between machine-generated and authentic images. This task is rendered feasible through the identification of subtle, yet machine-detectable, patterns unique to images generated by Generative Adversarial Networks (GANs), as evidenced in recent research (Marra et al., 2019; Afchar et al., 2018; Güera & Delp, 2018; Yu et al., 2019). Furthermore, the detection of deep fakes is enhanced by analyzing inconsistencies in the frequency domain or texture representation between authentic and fabricated images, as indicated in recent studies (Zhang et al., 2019; Durall et al., 2020; Liu et al., 2020).

**Authorship Identification for Natural Language Generation Models**

Likewise, content generated by text generation models is increasingly vulnerable to various forms of misuse, including the spread of misinformation and the training of surrogate models (Wallace et al., 2020; Xu et al., 2022). Consequently, a growing interest has been in protecting the authorship (or IP) of text generation models or detecting machine-generated text.

A straightforward solution is to incorporate watermarks into the generated text. However, unlike images, textual information is composed of discrete tokens, making the watermarking process for text a difficult endeavor due to the potential for inadvertent alternation that can change its semantic meaning (Katzenbeisser & Petitolas, 2000). One solution to preserve semantic integrity during watermarking involves synonym substitution (Topkara et al., 2006; Chang & Clark, 2014; He et al., 2022a). Nevertheless, the simplistic approach to synonym substitution is vulnerable to detection through statistical analyses. In response, He et al. (2022b) have proposed a conditional synonym substitution method to enhance both stealthiness and robustness of substitution-based watermarks. Moreover, Venugopal et al. (2011) adopted bit representation to encode semantically similar sentences, enabling the selection of watermarked sentences through bit manipulation.

The previously discussed methods have been centered on applying watermarks through post-editing. However, with the emergence of LLMs, there has been a significant shift towards developing watermarks tailored for LLMs to identify machine-generated text. A notable approach in this area employs biased sampling strategies to alter token distribution at each generation step, favoring tokens from specific pre-defined categories (Kirchenbauer et al., 2023a;b; Zhao et al., 2023a). Despite their innovation, these methods face vulnerabilities to "rewriting" attacks, where watermarked sentences are paraphrased either automatically or manually, thus challenging the identification of original authorship (Christ et al., 2023). To address this issue, Kuditipudi et al. (2023) propose a robust approach that maps a sequence of random numbers generated from a randomized watermark key to the outputs of a language model. This technique maintains the watermark's detectability, notwithstanding text alterations such as substitutions, insertions, or deletions.

Interest in post-hoc detection has surged as a complementary measure to watermarking. This trend is driven by the fact that developers often keep the details of their watermarking algorithms confidential to prevent them from being compromised if leaked. Depending on whether machine-generated or human-authored text samples are available, one can utilize either zero-shot or training-based detection methods. Zero-shot

detection relies on the premise that language model-generated texts inherently contain detectable markers, identifiable through analysis techniques such as perplexity (Idnay et al., 2022; Tian, 2023) or average per-token log probability (Solaiman et al., 2019a; Mitchell et al., 2023). The training-based approaches leverage features from pre-trained language models to distinguish between machine-generated and human-authored texts (Solaiman et al., 2019a; Bakhtin et al., 2019; Jawahar et al., 2020; Chen et al., 2023). However, these approaches yield a binary outcome, classifying texts as machine-generated or human-generated. Instead, our approach can determine the origin of text generated by any LLMs, moving beyond binary outcomes.

## 3 Background

The most recent advanced large generative models usually support two modes for generating outputs:

**Prompt-based Generation (*Generation* mode $\langle g \rangle$):** The authentic generator $\mathcal{G}$ produces outputs $\boldsymbol{x}_a$ conditioned on the given prompts $\boldsymbol{x}_p$, expressed as

$$\boldsymbol{x}_a = \mathcal{G}^{\langle g \rangle}(\boldsymbol{x}_p). \tag{1}$$

The selection of prompt inputs, denoted as $\boldsymbol{x}_p$ depends on the modalities supported by the generative models. These can vary widely, ranging from textual descriptions to images.

**Paraphrasing Content (*Paraphrasing* mode $\langle p \rangle$):** The generators can also "paraphrase" the intended outputs, including texts or images, by reconstructing the content,

$$\boldsymbol{x}_a^{\langle \text{new} \rangle} = \mathcal{G}^{\langle p \rangle}(\boldsymbol{x}_a^{\langle \text{old} \rangle}). \tag{2}$$

As examples of Natural Language Generation (NLG), we can use round-trip translation (Gaspari, 2006) or prompt the Large Language Model (LLM) to "paraphrase" a sentence. As an image generation (IG) example, given a generated image $\boldsymbol{x}_a$, the "paraphrasing" process will be (1) rebuilding partial images of the randomly masked region $M[t]$ in the original image $\boldsymbol{x}_a^{\langle \text{old} \rangle}$, *i.e.*, $\boldsymbol{x}_a^{\langle \text{new} \rangle}[t] = \mathcal{G}(\boldsymbol{x}_a^{\langle \text{old} \rangle}, M[t])$ (von Platen et al., 2022), and (2) merge these rebuilt portions as a whole image as the "paraphrased" output,

$$\boldsymbol{x}_a^{\langle \text{new} \rangle} = \text{Merge}(\{\boldsymbol{x}_a^{\langle \text{new} \rangle}[t]\}_{t=1}^T). \tag{3}$$

This iterative re-generation uses prior outputs as inputs and keeps their modality consistent. $x_a^{\langle 0 \rangle}$ is the initial output from the prompt-based generation.

Our method will use *Prompt-based Generation* mode for initial output generation and *Paraphrasing Content* mode for both (1) iteratively polishing the generated content and (2) authorship verification.

## 4 Methodology

Our research primarily focuses on the threat posed by malicious users who misuse generated content without authorization. Specifically, we delve into cases where the owner of authentic generative models, referred to as $\mathcal{G}_a$, allows access to their models in a black-box fashion, permitting external queries to their APIs for content generation. This black-box setting is consistent with the current application practices of Large Foundation Models. Hereby, unscrupulous users could take advantage of the generated content, $\boldsymbol{x}_a$, while disregarding the legitimate creator's license for their own profits. To address this issue, API providers can actively monitor the characteristics of publicly available data to identify potential cases of plagiarism or misuse. This can be accomplished by applying the re-generation and measuring the corresponding edit distance in the verification stage, as described in Section 4.2. To enhance verification accuracy, the authentic model $\mathcal{G}_a$ can employ an iterative re-generation approach to boost the fingerprinting signal, as introduced and proved in Section 4.3. If there are suspicions of plagiarism, the company can initiate legal proceedings against the alleged plagiarist through a third-party arbitration, following the verification protocol (see Algorithm 2) on the output with iterative re-generation (see Algorithm 1), as demonstrated in Section 4.1. The motivation and intuition of the approach proposed in Section 4.1 are explained in Sections 4.2 and 4.3.

---

**Algorithm 1** *Generation* Algorithm for *Stage I.*

---

**Input:** Prompt input $\boldsymbol{x}_p$ for generation (text or image) and the number of iterations $K$. $\mathcal{G}^{\langle g \rangle}(\cdot)$ and $\mathcal{G}^{\langle p \rangle}(\cdot)$ indicate the *generation* and *paraphrasing* modes of the authentic generative model respectively.
**Output:** The generated content (text or image) with implicit fingerprints.

1: $\boldsymbol{x}_a^{\langle 0 \rangle} \leftarrow \mathcal{G}^{\langle g \rangle}(\boldsymbol{x}_p)$       ▷ Initial generation from prompts $\boldsymbol{x}_p$ using *generation* mode $\mathcal{G}^{\langle g \rangle}$.
2: **for** $k \leftarrow 1$ to $K$ **do**              ▷ Iterate $K$ steps.
3:    $\boldsymbol{x}_a^{\langle k \rangle} \leftarrow \mathcal{G}^{\langle p \rangle}(\boldsymbol{x}_a^{\langle k-1 \rangle})$      ▷ Re-generation using *paraphrasing* mode with consistent types.
4: **end for**
5: **return** $\boldsymbol{x}_a^{\langle K \rangle}$

---

**Algorithm 2** *Verification* Algorithm for *Stage II.*

---

**Input:** Data sample $\boldsymbol{x}_a^{\langle K \rangle}$ generated by authentic generator $\mathcal{G}_a$ and another generator for contrasting $\mathcal{G}_c$. The threshold $\delta$ for confident authentication.
**Output:** Unauthorized usage according to a contrasting generator.

1: $\boldsymbol{y}_a \leftarrow \mathcal{G}_a^{\langle p \rangle}(\boldsymbol{x}_a^{\langle K \rangle})$       ▷ Regenerate data by model $\mathcal{G}_a^{\langle p \rangle}$ in *paraphrasing* mode.
2: $\boldsymbol{y}_c \leftarrow \mathcal{G}_c^{\langle p \rangle}(\boldsymbol{x}_a^{\langle K \rangle})$       ▷ Regenerate data by model $\mathcal{G}_c^{\langle p \rangle}$ in *paraphrasing* mode.
3: $r \leftarrow \mathbb{D}(\boldsymbol{y}_c, \boldsymbol{x}_a^{\langle K \rangle}) / \mathbb{D}(\boldsymbol{y}_a, \boldsymbol{x}_a^{\langle K \rangle})$      ▷ Calculate the exceeding distance ratio.
4: **return** $r > 1 + \delta$

---

## 4.1 Data Generation and Verification Protocol

The defense framework is comprised of two key components, responding to the *Generation* and *Verification* Stages in Figure 1:

(1) **Iterative Generation** (Algorithm 1), which progressively enhances the fingerprint signal in the generated outputs. Initially, a prompt input $\boldsymbol{x}_p$ is used to generate the first output $\boldsymbol{x}_a^{\langle 0 \rangle}$ using the generative model $\mathcal{G}$ in *generation* mode $\mathcal{G}^{\langle g \rangle}$. This output is then iteratively re-generated for $K$ steps using the same model but in *paraphrasing* mode, $\mathcal{G}^{\langle p \rangle}$, ensuring the type consistency of the content. Each iteration aims to reinforce the unique fingerprint of the authentic model in the generated content.

(2) **Verification** (Algorithm 2) is responsible for confirming the authorship of the data sample through a one-step re-generation process using both authentic model $\mathcal{G}_a$ and suspected contrasting model $\mathcal{G}_c$ in *paraphrasing* mode. The authentic and contrasting samples are then distinguished by their ratio with a confidence margin $\delta > 0$.

Our verification protocol takes inspiration from human artists' capability to reproduce their work that closely resembles the original. Similarly, our process of iterative re-generation mirrors the way artists refine their unique writing or painting styles. Moreover, the distinctive 'style' of the generative model becomes increasingly pronounced with each re-generation cycle, as it converges to a more 'stable' output. This process is formulated by the iterative functions and their tendency to converge to fixed points is proved in Theorem 2.

## 4.2 Authorship Verification through Re-Generation Distance

An authentic generative model $\mathcal{G}_a$ that aims to distinguish between the data samples it generated, denoted as $\boldsymbol{x}_a$, with the benign samples $\boldsymbol{x}_c$ generated by other models $\mathcal{G}_c$ for contrast. Unless stated otherwise, the generators $\mathcal{G}$ operate in *paraphrasing* mode in the following discussion, ensuring a uniform output modality during the re-generation process. To verify the data, the authentic model (1) re-generates the data $\mathcal{G}_a(\boldsymbol{x})$ and (2) evaluates the distance between the original sample and the re-generated sample, defined as $d(\boldsymbol{x}, \mathcal{G}) \triangleq \mathbb{D}(\mathcal{G}(\boldsymbol{x}), \boldsymbol{x})$. In essence, samples produced by the authentic model are expected to exhibit lower 'self-edit' distance, as they share the same generative model $\mathcal{G}_a$, which uses identical internal knowledge, such as

writing or painting styles, effectively serving as model fingerprints. In mathematical terms, we have

$$\mathbb{D}(\mathcal{G}_a(\boldsymbol{x}_a), \boldsymbol{x}_a) < \mathbb{D}(\mathcal{G}_a(\boldsymbol{x}_c), \boldsymbol{x}_c), \tag{4}$$

$$i.e., \; d(\boldsymbol{x}_a, \mathcal{G}_a) < d(\boldsymbol{x}_c, \mathcal{G}_a). \tag{5}$$

Consequently, models can identify the authentic samples generated by themselves by evaluating the 'self-edit' distance, which can be viewed as a specialized form of a classification function for discriminating the authentic and contrasting models. Additionally, the re-generation process and corresponding 'edit' distances can serve as explainable and comprehensible evidence to human judges.

### 4.3 Enhancing Fingerprint through Iterative Re-Generation

While we have claimed that the data samples generated through the vanilla generative process can be verified, there is no theoretical guarantee regarding the 'self-edit' distance for certifying these samples using the authentic model. To address this limitation, we introduce an iterative re-generation method that improves the fingerprinting capability of the authentic generative model, as it manages to reduce the 'self-edit' distances. This property will be utilized in verification, as the re-generation step in *Verification Stage* serves as the $(K + 1)$-th re-generation step after $K$ steps in the *Generation Stage* regarding the authentic model. Our theoretical foundation assumes the $L$-Lipschitz constant of the generative model $\mathcal{G}$ satisfying $L \in (0, 1)$. While measuring the exact Lipschitz constant for a deep generative AI model is challenging, especially for proprietary models, this property is empirically verifiable and we find it satisfied in many generative AI models, as presented in Appendix C.3.

**Definition 1 (The Fixed Points of a Lipschitz Continuous Function)** *Given a multi-variable function $f : \mathbb{R}^m \to \mathbb{R}^m$ is $L$-Lipschitz continuous,* i.e., $\|f(\boldsymbol{x}) - f(\boldsymbol{y})\| \leq L \cdot \|\boldsymbol{x} - \boldsymbol{y}\|$, where $L \in (0, 1)$. *For any initial point $\boldsymbol{x}^{\langle 0 \rangle}$, the sequence $\{\boldsymbol{x}^{\langle k \rangle}\}_{k=0}^{\infty}$ is acquired by the recursion $\boldsymbol{x}^{\langle k+1 \rangle} = f(\boldsymbol{x}^{\langle k \rangle})$. The sequence converges to a fixed point $\boldsymbol{x}^*$, where $\boldsymbol{x}^* = f(\boldsymbol{x}^*)$.*

**Theorem 2 (The Convergence of Step Distance for $k$-th Re-generation)** *Assuming that the Lipschitz constant $L$ of function $f$, which can be presented as a generative model, the distance between the input and output of the $K$-th iteration is bounded by*

$$\|\boldsymbol{x}^{\langle K+1 \rangle} - \boldsymbol{x}^{\langle K \rangle}\| \leq L^K \cdot \|\boldsymbol{x}^{\langle 1 \rangle} - \boldsymbol{x}^{\langle 0 \rangle}\|, \tag{6}$$

*and the distance converges to 0 given $L \in (0, 1)$.*[1]

**Proof** We apply $L$-Lipschitz continuous property recursively,

$$
\begin{aligned}
\|\boldsymbol{x}^{\langle K+1 \rangle} - \boldsymbol{x}^{\langle K \rangle}\| &= \|f(\boldsymbol{x}^{\langle K \rangle}) - f(\boldsymbol{x}^{\langle K-1 \rangle})\| \\
&\leq L \cdot \|\boldsymbol{x}^{\langle K \rangle} - \boldsymbol{x}^{\langle K-1 \rangle}\| = L \cdot \|f(\boldsymbol{x}^{\langle K-1 \rangle}) - f(\boldsymbol{x}^{\langle K-2 \rangle})\| \\
&\leq L^2 \cdot \|\boldsymbol{x}^{\langle K-1 \rangle} - \boldsymbol{x}^{\langle K-2 \rangle}\| \leq \cdots \\
&\leq L^k \cdot \|\boldsymbol{x}^{\langle 1 \rangle} - \boldsymbol{x}^{\langle 0 \rangle}\|.
\end{aligned}
\tag{7}
$$

∎

The theory can be extended to distance metrics in Banach space.

**Theorem 3 (Banach Fixed-Point Theorem)** *(Banach, 1922; Ciesielski, 2007) Let $(\mathcal{X}, \mathbb{D})$ be a complete metric space, and $f : \mathcal{X} \to \mathcal{X}$ is a contraction mapping, if there exists $L \in (0, 1)$ such that for all $\boldsymbol{x}, \boldsymbol{y} \in \mathcal{X}$*

$$\mathbb{D}(f(\boldsymbol{x}), f(\boldsymbol{y})) \leq L \cdot \mathbb{D}(\boldsymbol{x}, \boldsymbol{y}). \tag{8}$$

*Then, we have the following convergence of the distance sequence similar to Theorem 2,*

$$\mathbb{D}(\boldsymbol{x}^{\langle K+1 \rangle} - \boldsymbol{x}^{\langle K \rangle}) \leq L^K \cdot \mathbb{D}(\boldsymbol{x}^{\langle 1 \rangle} - \boldsymbol{x}^{\langle 0 \rangle}). \tag{9}$$

---

[1]We estimated $L$ for Stable Diffusion models with results presented in Appendix C.3.

# 5 Experiments

This section aims to demonstrate the efficacy of our re-generation framework of authorship authentication on generated text and image separately. For both Natural Language Generation (NLG) and Image Generation (IG) scenarios, in Sections 5.2 and 5.3, we first generate the initial intended data samples followed by several steps of 'paraphrasing' (for NLG) or 'inpainting' (for IG) as re-generation, detailed in Algorithm 1. Then, we test the properties of these samples by three series of experiments.

1. **Distance Convergence**: We verify the convergence of the distance between all one-step re-generations $k \in [1..K]$.

2. **Discrepancy**: We illustrate the discrepancy between the distances by the authentic generative models and the 'suspected' other models for contrast.

3. **Verification**: We report the verification performance using precision and recall of the identified samples by the authentic models vs those from other contrasting models.

## 5.1 Experimental Setup

Despite falling under the same framework for authorship verification, Natural Language Generation (NLG) and Image Generation (IG) use distinct generative models and varying metrics for measuring distance.

**Generative Models.** For text generation, we consider four generative models: 1) **M2M** (Fan et al., 2021): a multilingual encoder-decoder model trained for many-to-many multilingual translation; 2) **mBART-large-50** (Tang et al., 2021): a model fine-tuned on mBART for multilingual machine translation between any pair of 50 languages; **Cohere**: a large language model developed by Cohere; 4) **GPT3.5-turbo** (version: 0613): a chat-based GPT3.5 model developed by OpenAI.[2]

For image generation, we examine five primary generative models based on the Stable Diffusion (SD) architecture (Rombach et al., 2022). All models are trained on a subset of the LAION-2B dataset (Schuhmann et al., 2022) consisting of CLIP-filtered image-text pairs. These models are: 1) **SDv2.1**; 2) **SDv2**; 3) **SDv2.1 Base**; 4) **SDXLv1.0** (Podell et al., 2023), which distinguishes itself by employing an ensemble of experts pipeline for latent diffusion, where the base model first generates noisy latent that is refined using a specialized denoising model 5) **SDXL Base0.9** (Podell et al., 2023).[3] More information on model architecture and training, and the quality of re-generated images is demonstrated in Appendix C.1.

**Distance Metrics.** To gauge the similarity between generated outputs, we employ three popular similarity metrics for NLG experiments, 1) **BLEU** calculates the precision of the overlap of various n-grams (usually 1 to 4) between the candidate and the reference sentences (Papineni et al., 2002); 2) **ROUGE-L** calculates the F-1 score of longest common subsequence between the candidate and reference sentences (Lin, 2004) and 3) **BERTScore** computes the similarity between candidate and reference tokens using cosine similarity on their embeddings. Then, the token-level scores are aggregated to produce a single score for the whole text (Zhang et al., 2020). For IG experiments, we consider the following distance metrics: 1) **CLIP Cosine Distance** measures the semantic similarity of two images using pre-trained CLIP image-text models (Radford et al., 2021); 2) **LPIPS** compares perceptual style differences between two images (Zhang et al., 2018); 3) **Mean Squared Error (MSE)** serves as a pixel-level metric that compares all raw pixel values in the image pair; 4) **Structural Similarity Index (SSIM)** assesses image degradation based on luminance, contrast, and structure (Wang et al., 2004). We transform all similarity scores $s \in [0, 1]$ to distances $d$ by $d = 1 - s$.

Note that we gauge the success of our approach based on the satisfactory performance of any of these metrics when applied to the verification of generative models.

---

[2]We have also studied GPT4 and present its corresponding results in the Appendix D.2.
[3]Additionally, we explored other variants, including **SDv1.5**, **SDv1.4**, and **Nota-AI's BK-SDM Base**. The corresponding experiments and results are provided in Appendix C.1.

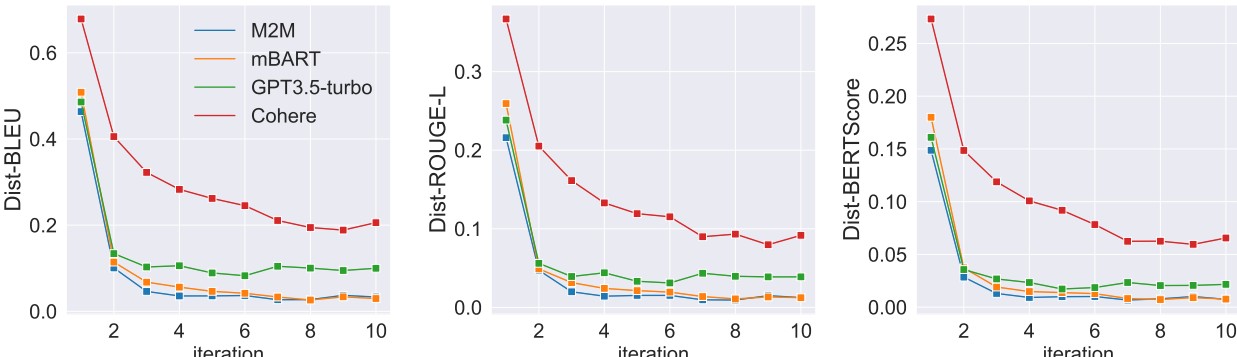

Figure 2: The convergence analysis of the distances in iterations based on various metrics on the re-generated text of 200 samples from in-house datasets.

## 5.2 Natural Language Generation Experiments

In the study of text generation, the primary objective is to produce sentences that fulfill specific requirements, including translation, paraphrasing, summarization, *etc.*. Then, one can re-generate intended output sentences via 'paraphrasing' them. This study focuses on machine translation (specifically from French to English) as the primary generation task.[4] Considering that M2M and mBART are restricted to machine translation tasks, we utilize round-trip translation as a way for paraphrasing. Specifically, for each round-trip translation step, an English input sentence is first translated into French and then translated back to English. We repeat this process multiple times to observe the change of the 'edit' distances. To reproduce the primary experiment, *i.e.,* Fr-En translation, using GPT3.5-turbo and Cohere, we perform a zero-shot prompting using the following prompts:

- Translate to English: *You are a professional translator. You should translate the following sentence to English and output the final result only: {**INPUT**}*

- Translate to French: *You are a professional translator. You should translate the following sentence to French and output the final result only: {**INPUT**}*

Text generative models have been known to exhibit outstanding performance when test data inadvertently overlaps with pre-training data, a phenomenon referred to as data contamination (Magar & Schwartz, 2022). To address this potential issue, we sample 200 sentences from the in-house data as the starting point for each model. To mitigate biases arising from varied sampling strategies, we set the temperature and top p to 0.7 and 0.95 for all NLG experiments.

**Distance Convergence.** The dynamics of the distance change across three metrics and various generative models are depicted in Figure 2. We observe a remarkable distance reduction between the first and second iterations in all settings. Subsequently, the changes in distance between consecutive iterations exhibit a diminishing pattern, tending to converge after approximately 5-7 rounds of re-generation. These observed trends are consistent with the Fixed-Point Theorem as elaborated in Section 4.3.

**Discrepancy.** In our previous study of iterative re-generation using the same model, we observe the convergence of 'edit' distances which can be utilized to distinguish the authentic model from its counterparts. As delineated in Section 4.2, for each French sentence $x$ from the corpus, we apply the prompt-mode of authentic model to yield $x_a$ via a translation. Both authentic and contrasting models are considered to perform a one-step re-generation of $x_a$ to obtain $y$ for verification. We measure the distance between $x_a$ and $y$ (including $y_a$ and $y_c$). As illustrated in Figure 3, the density distribution associated with the authentic model demonstrates a noticeable divergence from those of models for contrast, thus affirming the

---

[4]In addition to machine translation, we examine paraphrasing and summarization as the generation tasks in Section 5.4

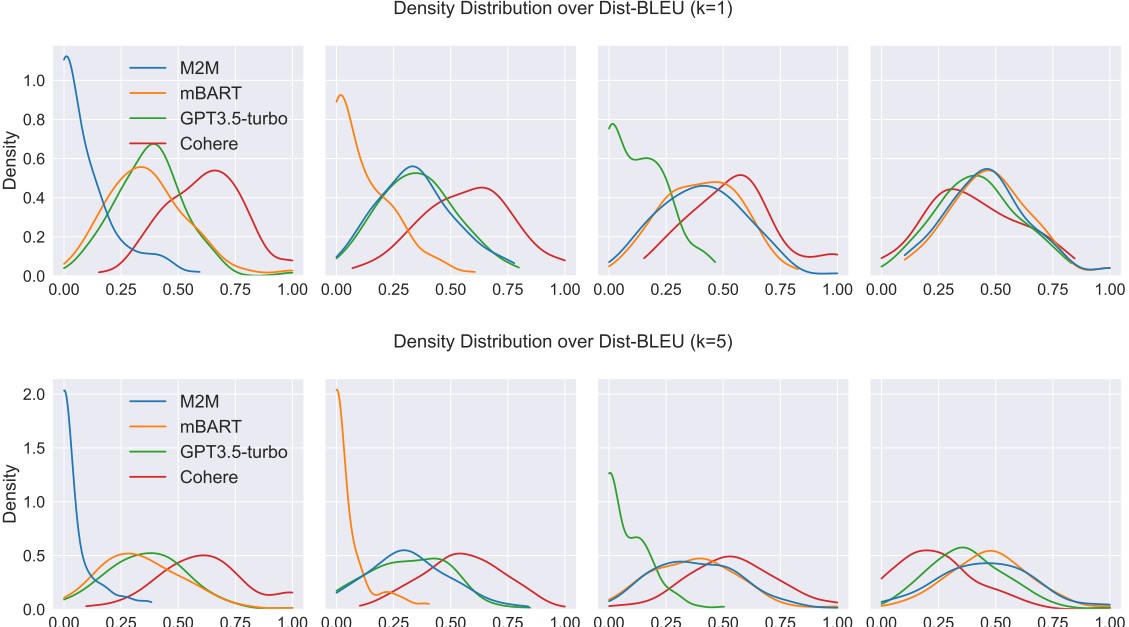

Figure 3: Density distribution of one-step re-generation among four text generation models, where the input to the one-step re-generation is the $k$-th iteration from the authentic models. The authentic models from left to right are: 1) M2M, 2) mBART, 3) GPT3.5-turbo, and 4) Cohere.

hypothesis discussed in Section 4.2. This pattern also holds when using ROUGE-L and BERTScore for distance measures, as evidenced by their respective density distributions in Appendix D.1.

We observe the difficulty of distinguishing Cohere from other models merely based on the outputs from the first iteration, *i.e.,* without re-generation. We attribute the distinctive behavior of Cohere to its relatively slower convergence than other models.[5] However, as depicted in Figure 2, the distances observed in subsequent iterations of the same model experience a substantial reduction. This implies the re-generated outputs in the Generation Stage possess enhanced fingerprints. In particular, we can derive $x_a^{\langle k \rangle}$ from 5 iterations of $\mathcal{G}_a(\cdot)$. Figure 3 shows that increasing $k$ enhances the differentiation of the authentic model's density distribution compared to other models. This distinction becomes evident even in the case of Cohere, which remains indistinguishable at $k = 1$. This enhancement for Cohere is also corroborated in the verification experiments in the next paragraph.

**Verification.** We employ Algorithm 2 to ascertain if a given sentence originates from the authentic models. The determination hinges on the threshold parameter, $\delta$. Thus, we designate M2M as the authentic model, while using mBART, Cohere, and GPT3.5-turbo as contrasting models to determine the optimal value of $\delta$. As indicated in Table 1, a threshold of 0.05 certifies that the three contrasting models can validate that more than 93% of the samples are derived from M2M. As anticipated, an increase in the value of $\delta$ augments the stringency criteria, resulting in a reduction in verification precision. Therefore, we fix $\delta$ at 0.05 for ensuing evaluations unless stated otherwise.

As indicated in Table 2, our approach undertakes the validation of authorship, yielding a precision exceeding 85% and a recall rate surpassing 88% across M2M, mBART, and GPT3.5-turbo. Moreover, both precision and recall metrics improve with additional re-generation iterations. Although with a slower convergence speed on Cohere, the verification performance on its samples improved significantly given more re-generation iterations, see Cohere lines in Table 2.

---

[5]The underlying reason behind the slow convergence for Cohere is investigated in Appendix D.3.1.

Table 1: The precision of differentiating contrast models ($\mathcal{G}_c$, **mBART**, **Cohere**, and **GPT3.5-turbo**) from the authentic model ($\mathcal{G}_a$, **M2M**) using various $\delta \in \{0.05, 0.1, 0.2, 0.4\}$, based on BLUE, ROUGE-L, and BERTScore, respectively.

| $\delta$ | mBART | | | Cohere | | | GPT3.5-turbo | | |
|---|---|---|---|---|---|---|---|---|---|
| | BLEU | ROUGE | BERT | BLEU | ROUGE | BERT | BLEU | ROUGE | BERT |
| 0.05 | 94.0 | 92.0 | 91.0 | 99.0 | 99.0 | 99.0 | 93.0 | 89.0 | 92.0 |
| 0.10 | 91.0 | 92.0 | 90.0 | 98.0 | 99.0 | 99.0 | 92.0 | 88.0 | 92.0 |
| 0.20 | 91.0 | 92.0 | 90.0 | 98.0 | 99.0 | 99.0 | 91.0 | 87.0 | 90.0 |
| 0.40 | 88.0 | 88.0 | 85.0 | 95.0 | 97.0 | 99.0 | 87.0 | 85.0 | 85.0 |

Table 2: The precision and recall of verifying the authentic models ($\mathcal{G}_a$) using different contrasting models ($\mathcal{G}_c$).

| $\mathcal{G}_c$ / $\mathcal{G}_a$ | M2M | | mBART | | Cohere | | GPT3.5-turbo | |
|---|---|---|---|---|---|---|---|---|
| | Precision ↑ | Recall ↑ | Precision ↑ | Recall ↑ | Precision ↑ | Recall ↑ | Precision ↑ | Recall ↑ |
| (a) $k = 1$ | | | | | | | | |
| M2M | - | - | 94.0 | 98.0 | 99.0 | 99.0 | 93.0 | 95.0 |
| mBART | 85.0 | 89.0 | - | - | 100.0 | 100.0 | 89.0 | 92.0 |
| Cohere | 60.0 | 64.0 | 63.0 | 57.0 | - | - | 65.0 | 51.0 |
| GPT3.5-turbo | 87.0 | 92.0 | 90.0 | 92.0 | 99.0 | 99.0 | - | - |
| (b) $k = 3$ | | | | | | | | |
| M2M | - | - | 94.0 | 99.0 | 100.0 | 100.0 | 95.0 | 97.0 |
| mBART | 85.0 | 93.0 | - | - | 100.0 | 100.0 | 90.0 | 96.0 |
| Cohere | 75.0 | 83.0 | 73.0 | 76.0 | - | - | 63.0 | 69.0 |
| GPT3.5-turbo | 89.0 | 92.0 | 93.0 | 95.0 | 97.0 | 98.0 | - | - |
| (c) $k = 5$ | | | | | | | | |
| M2M | - | - | 94.0 | 98.0 | 98.0 | 99.0 | 95.0 | 99.0 |
| mBART | 91.0 | 96.0 | - | - | 100.0 | 100.0 | 88.0 | 94.0 |
| Cohere | 71.0 | 81.0 | 83.0 | 87.0 | - | - | 69.0 | 73.0 |
| GPT3.5-turbo | 90.0 | 94.0 | 94.0 | 98.0 | 95.0 | 97.0 | - | - |

### 5.3 Image Generation Experiments

The main objective of the image generation task is to produce an image given a text prompt using a generative model, *a.k.a prompt-mode*. To generate the initial proposal of images, we sample 200 captions as prompts each from **MS-COCO dataset (COCO)** (Lin et al., 2014) and **Polo Club Diffusion DB dataset (POLO)** (Wang et al., 2022), then we generate initial images $\boldsymbol{x}^{\langle 0 \rangle}$ corresponding to the prompts using all assigned models. For re-generation, we consider two settings: (1) *watermarking* images through inpainting a pre-defined sub-regions and (2) *fingerprinting* enhancement through full image re-generation. The subsequent two paragraphs detail the methodologies for each setting.

**Watermarking through Sub-Region Inpainting** When re-generating authentic image $\boldsymbol{x}_a$, we first mask a sub-region, *i.e.,* $1/N$ ($N = 10$ in our experiments) of its pixels with fixed positions as the watermark pattern. Then, all pixels in the masked regions are re-generated by inpainting using a generative model. Similar to text generation, the inpainting step is iterated multiple times, following Algorithm 1. The comprehensive description of the watermarking procedure is provided in Appendix B.1.

**Fingerprinting through Full-Image Re-generation** In the re-generation setting, we first split all possible mask positions into N ($N = 8$ in our experiments) non-overlapped sets, coined *segments*, each with $1/N$ pixel positions. We parallly reconstruct each segment based on the rest $(N-1)/N$ pixels of the image as the context in generation. Then, we reassemble these $N$ independent segments into a new full image $\boldsymbol{y}$. When analyzing reconstructions of $\boldsymbol{y}_a = \mathcal{G}_a(\boldsymbol{x}_a)$ by contrasting models $\boldsymbol{y}_c = \mathcal{G}_c(\boldsymbol{x}_a)$, there is an expectation that models may exhibit variations in painting masked regions due to their inherent biases.

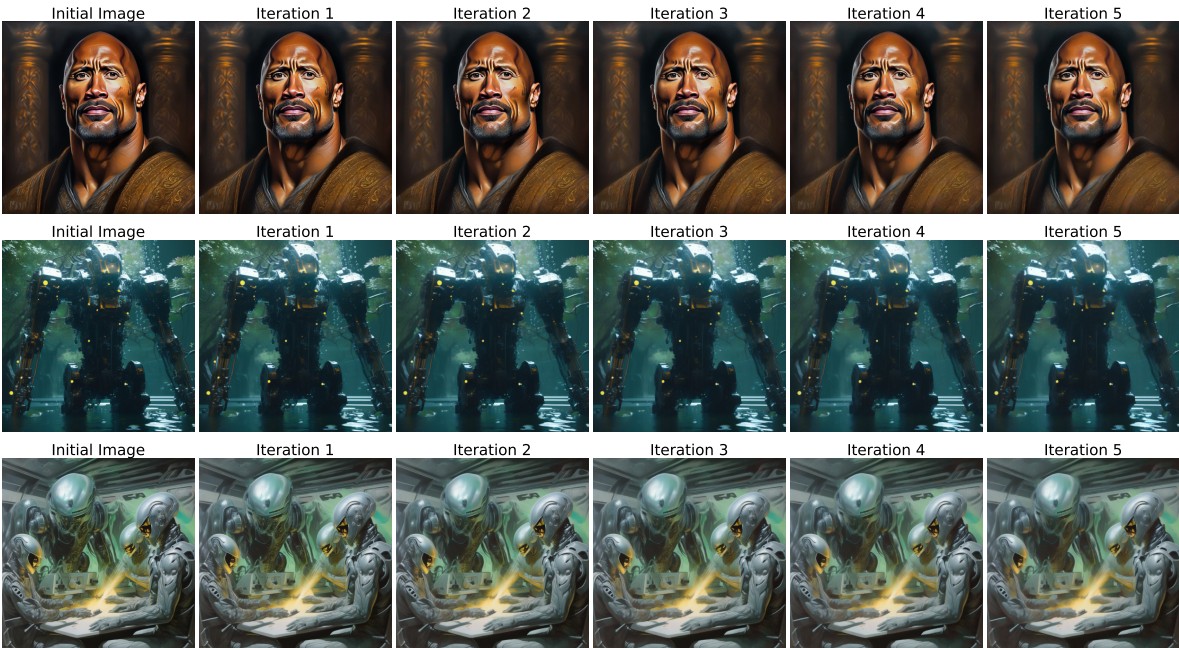

Figure 4: The quality of fingerprinted images over multiple iterations of full-image re-generation using SDXL 0.9B.

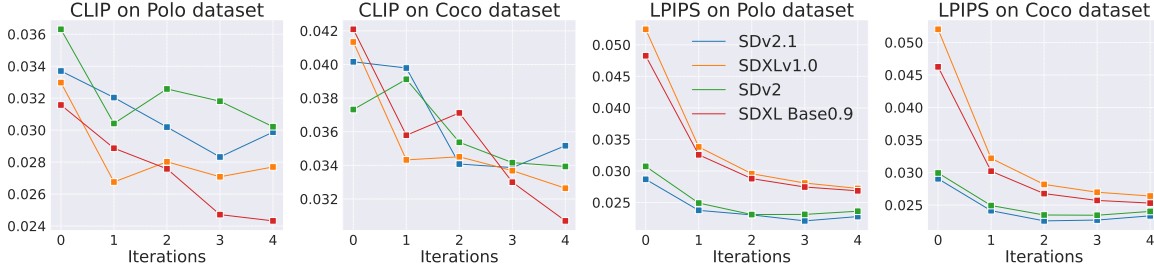

Figure 5: The convergence analysis of the distances in iterations based on CLIPS and LPIPS metrics on re-generated images of 200 samples on Coco and Polo datasets.

In this study, to generate the samples from the authentic models, each model first generates 200 initial images $x^{\langle 0 \rangle}$ given the captions, which are then re-generated over one to five iterations using 'inpainting' approaches. In the verification stage, by comparing $y_a$ and $y_c$ to the original $x_a$, we aim to identify subtle model fingerprints based on the 'edit-distance' from the original data to re-generated content. We use LPIPS and CLIP similarity between image pairs to estimate their difference.[6] Additionally, the consistency of the image quality during re-generated is demonstrated in Figure 4 and a detailed discussion is provided in Appendix D.3.2.

**Distance Convergence.** Similar to text re-generation experiments, the re-generation distance converges for both watermarking and fingerprinting settings, as illustrated in Figures 5 and 11, indicated by consistent downward trends across datasets and distance metrics. This implies that the re-generation process converges, with subsequent iterations yielding images that more closely resemble their predecessors. While our methodology proves efficacious in both scenarios, subsequent sections will prioritize the full-image regeneration setting as it is independent of the masks, which are considered confidential to the generative model. Details about the watermarking setting can be found in Appendix B.1.1.

---

[6]The results based on MSE and SSIM are also reported in Appendix B.2.1.

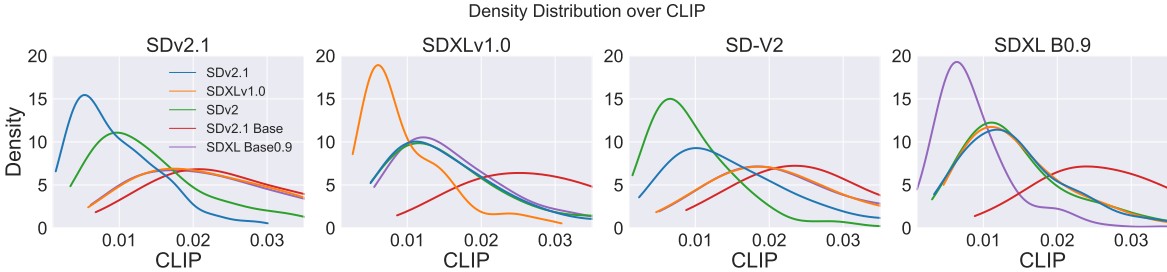

Figure 6: Density distribution of one-step re-generation among four image generation models on Polo Dataset. The authentic models from left to right are: 1) SD 2.1, 2) SDXL 1.0, 3) SD 2, 4) SDXL Base 0.9.

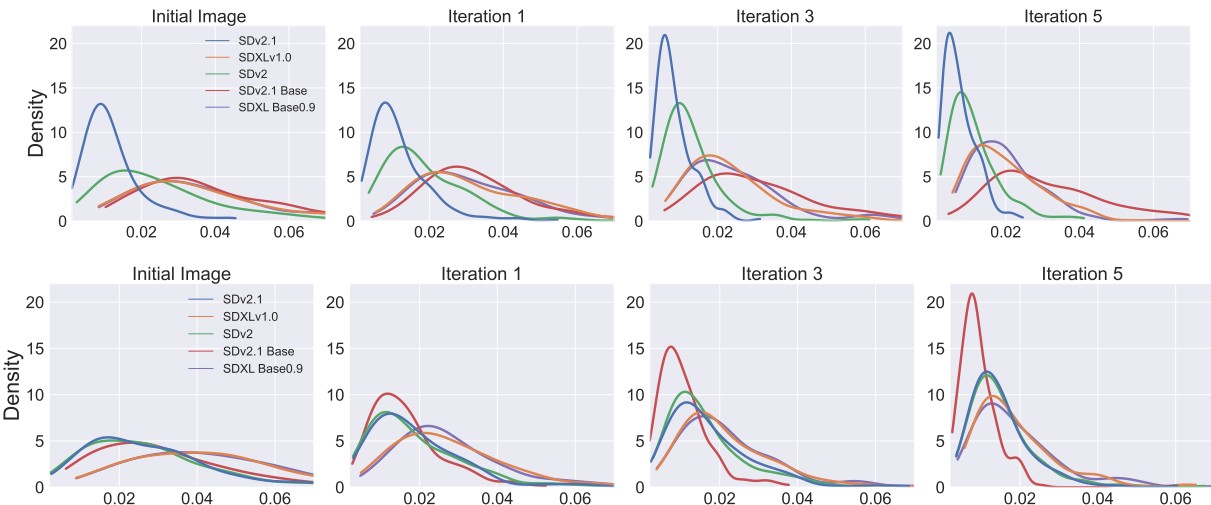

Figure 7: Verifying data generated by authentic $\mathcal{G}_a$ SDv2.1 and SDv 2.1 Base on Coco Dataset at various iterations using CLIP distance.

**Discrepancy.** In our study, we focus on the setting of iterative re-generation through inpainting and observe convergence that can reveal differences between models. Like the NLG section, for verification, both authentic and contrasting models perform an additional one-step re-generation of $\boldsymbol{x}_a$ to obtain $\boldsymbol{y}$ and the CLIP distance is measured between the outputs. As illustrated in Figure 6, the one-step re-generation density distribution of the authentic model predominantly peaks at lower values across all generative models. This indicates that authentic models can effectively distinguish their outputs from those of non-authentic models.

Then, we examine the effectiveness of 'inpainting' as re-generation for *Generation Stage.* As illustrated in Figure 7, re-generation can successfully enhance their fingerprints by $k = 5$ iterations. While most models, on average, may require fewer rounds of re-generation to deeply impart their fingerprints, with more rounds of re-generation, even older models' fingerprints can be successfully enhanced. Overall, the quality of the images is retained during re-generation, as showcased in Figure 4. At the same time, the strength of the fingerprints embedded in model outputs is reinforced over iterations, as illustrated in Figure 7.

**Verification.** In accordance with the text generation framework, we utilize Algorithm 2 to evaluate verification performance, selecting optimal parameters $k = 5$ and $\delta = 0.05$ based on the results of the text generation experiment. As demonstrated in Table 3, models reliably identify images from the authentic model, with precision often exceeding 80% and recall around 85%.

Table 3: The precision and recall of verifying the authentic models $\mathcal{G}_a$ using different contrasting models $\mathcal{G}_c$ on Polo Dataset, given $k = 5$ and $\delta = 0.05$.

| $\mathcal{G}_a$ \ $\mathcal{G}_c$ | SD v2.1 Precision ↑ | Recall ↑ | SD v2 Precision ↑ | Recall ↑ | SD v2.1B Precision ↑ | Recall ↑ | SDXL 0.9 Precision ↑ | Recall ↑ | SDXL 1.0 Precision ↑ | Recall ↑ |
|---|---|---|---|---|---|---|---|---|---|---|
| SD v2.1 | - | - | 80.0 | 86.0 | 99.5 | 99.5 | 98.5 | 99.5 | 99.0 | 100.0 |
| SD v2 | 77.5 | 81.5 | - | - | 99.0 | 100.0 | 96.5 | 97.0 | 95.5 | 97.5 |
| SD v2.1B | 81.5 | 83.5 | 82.5 | 86.5 | - | - | 88.5 | 92.0 | 89.5 | 91.0 |
| SDXL 0.9 | 97.5 | 97.5 | 96.5 | 98.0 | 99.5 | 99.5 | - | - | 92.0 | 93.5 |
| SDXL 1.0 | 95.5 | 96.0 | 92.5 | 95.5 | 100.0 | 100.0 | 94.5 | 96.0 | - | - |

## 5.4 Generalization and Robustness Evaluation

**Task Generalization** To demonstrate the generalization of our approach, we further examine two popular text generation tasks, paraphrasing and summarization, as the target generation tasks. The prompts for the prompt mode are:

- **Paraphrasing**: *You are a professional language facilitator. You should paraphrase the following sentence and output the final result only: {**INPUT**}*

- **Summarization**: *You are a professional language facilitator. You should summarize the following document using one sentence: {**INPUT**}*

We consider prompt-based paraphrasing for one-step re-generation (*i.e.,* paraphrasing mode), opting for a $k$ value of 5 due to its superior performance. As for backbone models for text generation tasks, we consider GPT3.5-turbo, Llama2-chat-7B (Touvron et al., 2023), and Mistral (Jiang et al., 2023a). Table 4 demonstrates that while paraphrasing and summarization tasks exhibit performance degradation compared to machine translation, our approach effectively distinguishes IPs among various models. This supports the overall effectiveness of our approach across a wide range of text-generation tasks.

Table 4: The precision of verifying the authentic models ($\mathcal{G}_a$) using different contrasting models ($\mathcal{G}_c$). The target tasks are paraphrasing and summarization.

| $\mathcal{G}_a$ \ $\mathcal{G}_c$ | GPT3.5-turbo Precision ↑ | Zephyr Precision ↑ | Mistral Precision ↑ |
|---|---|---|---|
| Paraphrasing | | | |
| GPT3.5-turbo | - | 75.0 | 69.0 |
| Zephyr | 84.0 | - | 69.0 |
| Mistral | 74.0 | 76.0 | - |
| Summarization | | | |
| GPT3.5-turbo | - | 92.0 | 68.0 |
| Zephyr | 88.0 | - | 67.0 |
| Mistral | 86.0 | 86.0 | - |

**Robustness Evaluation** To examine the robustness of our verification process, we corrupt the generated outputs using paraphrasing and perturbation attacks.

Our robustness assessment commenced with a paraphrasing attack targeting image generation. This experiment was designed to test the efficacy of our verification process in identifying plagiarized images. Specifically, we consider images that are originally generated by one authentic model (Model A) and then re-generated or paraphrased by another attack model (Model B), coined as *A_B*. We experiment with SD v2.1 as the authentic model *A* and SDXL 1.0 as the paraphrasing model *B*.

In verification, we re-generate these paraphrased *A_B* images using both the authentic and paraphrasing models to analyze the distance density distribution variations. Furthermore, we assess this paraphrasing attack at different rounds of generation $\boldsymbol{x}_a^{\langle K \rangle}$ where K is 1, 3, and 5 by the authentic model.

The results, represented in Figure 8, show comparisons between $\mathbb{D}(A\_B, \mathcal{G}_A(A\_B))$ and $\mathbb{D}(A\_B, \mathcal{G}_B(A\_B))$, indicating the distance between the paraphrased image (created by A then paraphrased by B) and re-generated images.

We observe a significant challenge in attributing authorship to the original creator model once the image is plagiarized (even at higher values of re-generation iteration K in Algorithm 1). The distance between the

paraphrased image to the authentic model and the paraphrasing model cannot be differentiated making it increasingly difficult to distinguish whether the SDv2.1 model or another paraphrasing model generated the image.

However, the results also reveal a noteworthy strength in the methodology. There is a discernible and consistently large distance when comparing paraphrased images to those generated by a contrast model (*i.e.,* an unrelated model, one which was neither the authentic nor the paraphrasing model). This finding is crucial as it suggests that despite the challenges in direct attribution, our approach can reliably detect when an external model, not involved in the image's original creation or its paraphrasing, is falsely claimed as the author. Therefore, this method could potentially be leveraged to establish joint authorship in scenarios where the paraphrased image is claimed by a third party, ensuring proper credit is given to the authentic and paraphrasing models involved in the image's creation.

We investigate perturbation attacks in both text and image generation tasks. For our NLP experiments, we randomly perturb $X\%$ of the words preceding the one-step re-generation procedure by substituting them with random words. The value of $X$ ranges from 10% to 50%. We designate GPT3.5-turbo as the authentic model. We set $k = 5$, due to its outperforming performance. As depicted in Table 5, as the perturbation rate increases, we observe a gradual decline in the verification performance of our method. In general, the efficacy of watermark verification remains reasonably positive. Nevertheless, we argue that perturbations in the range of 30% to 50% have

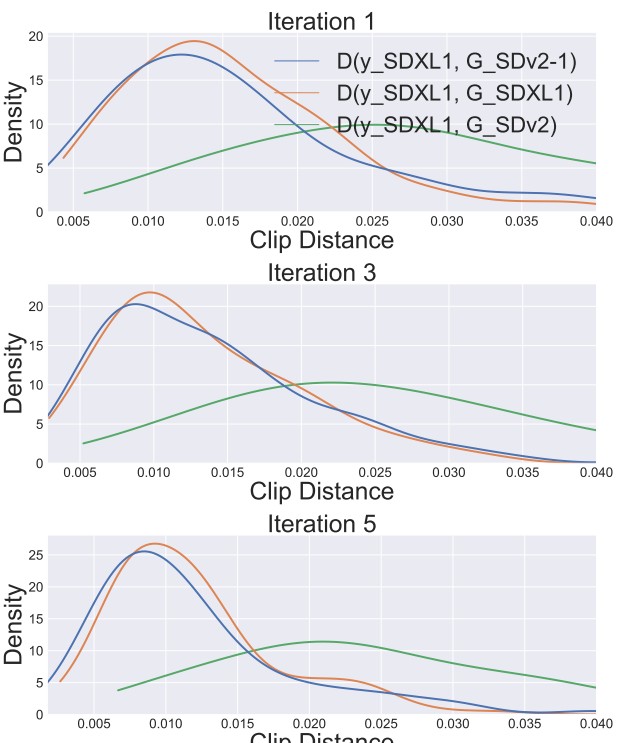

Figure 8: The variations in the distance density distribution of paraphrasing images through the authentic model and contrasting model. The paraphrased images are initially generated by SDv2.1 (authentic model) and then paraphrased by models SDXL 1.0 at different iterations K (1, 3 & 5) by the authentic model. Additionally, SDv2 acts as a contrasting model for the set of images which is produced by SDv2.1 and then paraphrased by SDXL 1.0.

a notable adverse impact on the quality of the generated texts, rendering them significantly less valuable for potential attackers.

Table 5: The evaluation of robustness evaluation for text generation with perturbed substitution when $k = 5$ using precision.

| Perturbation rates | M2M | mBART | Cohere |
|---|---|---|---|
| 0% | 90.0 | 94.0 | 95.0 |
| 10% | 91.0 | 91.0 | 97.0 |
| 20% | 84.0 | 84.0 | 95.0 |
| 30% | 84.0 | 74.0 | 94.0 |
| 40% | 79.0 | 75.0 | 93.0 |
| 50% | 69.0 | 65.0 | 92.0 |

Table 6: The robustness evaluation of image generation under brightness perturbation at Different Levels for Models SD v2, SD v2.1B, and SDXL 1.0 (k = 5).

| Perturbation rate | SD v2 | SD v2.1B | SDXL 1.0 |
|---|---|---|---|
| 0.00 | 76.0 | 97.5 | 94.0 |
| 1.01 | 35.0 | 97.0 | 85.0 |
| 1.02 | 41.5 | 91.0 | 87.0 |
| 1.05 | 34.5 | 94.0 | 59.5 |
| 1.10 | 35.0 | 93.0 | 44.0 |
| 1.50 | 23.5 | 91.5 | 41.5 |

Regarding image generation, we introduce two types of perturbations: Brightness alterations and Gaussian Noise. For brightness alterations, we modify the brightness of 10% of randomly chosen pixels. As the intensity of this perturbation increases, we observe a corresponding decrease in verification precision. Notably, as detailed in Table 6, the precision for the SDXL 1.0 model declines sharply but stabilizes when the perturbation rate reaches 1.02. Beyond this point, the model's robustness significantly deteriorated. In our second experiment, we applied Gaussian noise with a mean ($\mu = 0$) and standard deviation ($\sigma = 4$) to a random selection of image pixels. This process has a negligible effect on verification accuracy across different models, as shown in Table 7. Even with noise affecting up to 50% of the pixels, the models demonstrated a high precision range of 89-99%, underscoring their robustness to substantial random noise.

Table 7: The robustness evaluation for image generation under various levels of Gaussian Noise perturbation, where $\mu = 0$, $\sigma = 4$ at iteration $k = 5$.

| Perturbation rates | SD v2 | SD v2.1B | SDXL 1.0 |
|---|---|---|---|
| 0.00 | 76.0 | 97.5 | 94.0 |
| 0.01 | 82.0 | 94.5 | 82.0 |
| 0.05 | 100.0 | 98.5 | 91.0 |
| 0.10 | 97.0 | 99.0 | 95.0 |
| 0.20 | 99.5 | 100.0 | 93.5 |
| 0.50 | 99.0 | 99.0 | 89.5 |
| 1.00 | 99.5 | 99.0 | 66.5 |

**Distance Differentiation between Natural and AI-Generated Images** In order to assess the viability of our framework to detect plagiarism on naturally made images (*i.e.,* shot on camera), we re-generate natural images from the MS COCO 2017 Evaluation set (Lin et al., 2014) with previously used image generation models for one iteration. We compare the Clip distances between these natural images and their respective re-generations to the distance between the AI-generated images and their re-generation for one iteration by the corresponding models as presented in Figure 9. There is a clear distinction in the distance density distribution when AI-generated images are re-generated using AI our Generation Algorithm 1, the distance remains low whereas using a natural image for re-generation results in further distances even with the most advanced AI models.

Table 8: Precision, Recall, and F1 Score for different thresholds in classifying Human-Generated vs. AI-Generated images. The values are also translated into mean +/- $\tau \cdot$ std.

| Threshold | Precision | Recall | F1 Score | $\tau$ (Natural) | $\tau$ (AI) |
|---|---|---|---|---|---|
| 0.01 | 0.5797 | 1.000 | 0.7339 | -2.1825 | -0.6311 |
| 0.02 | 0.7418 | 1.000 | 0.8517 | -1.8827 | 0.0914 |
| 0.03 | 0.8701 | 0.991 | 0.9266 | -1.5829 | 0.8139 |
| 0.04 | 0.9280 | 0.942 | 0.9349 | -1.2831 | 1.5364 |
| 0.05 | 0.9572 | 0.850 | 0.9004 | -0.9833 | 2.2589 |
| 0.06 | 0.9716 | 0.719 | 0.8264 | -0.6835 | 2.9814 |
| 0.07 | 0.9804 | 0.600 | 0.7444 | -0.3837 | 3.7038 |

To quantify this distinction, we applied various thresholds to these distances and evaluated our method's performance. The results, shown in Table Table 8, illustrate the trade-offs between precision and recall at different thresholds. For example, at a threshold of 0.03, we achieved a high balance with a precision of 0.8701, recall of 0.991, and an F1 score of 0.9266, demonstrating robustness of the method. To facilitate threshold selection, we demonstrate the number of standard deviation from the mean CLIP distance for both *Natural* and *AI-generated* images, $\tau = (\text{threshold} - \text{mean})/\text{std}$. This standardization provides an interpretation of each threshold's position relative to the data distribution, enhancing the explainablility and reliability of the threshold selection.

## 6 Conclusion

In this work, we observe the intrinsic fingerprint in both text and vision generative models, which can be identified and verified by contrasting the re-generation of the suspicious data samples by authentic models and contrasting models. Furthermore, we propose iterative re-generation in the Generation Stage to enhance the fingerprints and provide a theoretical framework to ground the convergence of one-step re-generation distance by the authentic model. Our research paves the way towards a generalized authorship authentication for deep generative models without (1) modifying the original generative model, (2) post-processing to the generated outputs, or (3) an additional model for identity classification.

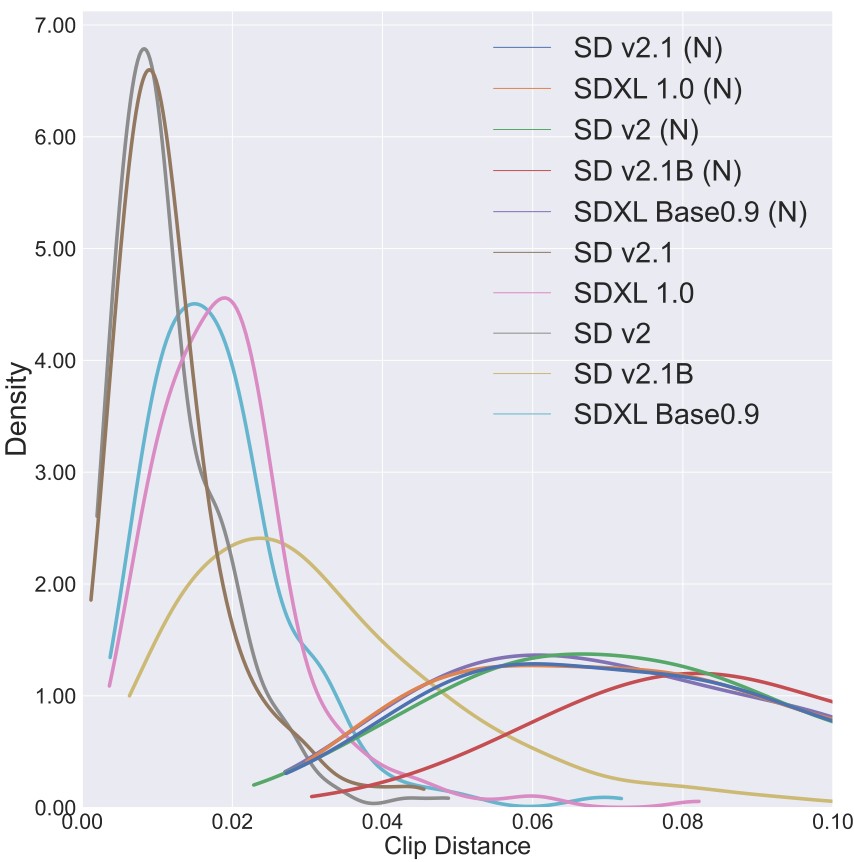

Figure 9: Comparison of Density Distribution of Re-generation Distances in Verification Stage, acquired by Natural and AI Image Re-generations.

## Limitations

Our approach, while effective in many scenarios, has several limitations. Firstly, it assumes that malicious users employ the authentic model and then directly utilize or disseminate them. However, users may modify the content to circumvent IP infringements, *e.g.,* involving other generative models. This poses a challenge for our method in protecting IP claims for altered content, which was also highlighted as a fundamental challenge in verifying AI-generated content (Jiang et al., 2023b). For image generation tasks, our approach relies on the inpainting methodology, which is native to the stable diffusion set of models (as described in C.1). As a result, we currently cannot empirically verify the effectiveness of our approach for other image generative models, such as GAN (Esser et al., 2021; Kang et al., 2023) and VAEs (Vahdat & Kautz, 2020). Nonetheless, considering that many of the state-of-the-art image generators are based on similar diffusion based architectures, it is reasonable to hypothesize that our method is applicable to them as well. Lastly, while our approach effectively differentiates between natural and AI-generated images, determining the optimal threshold for this differentiation requires careful balancing of precision and recall. This need for the hyperparemeter search highlights the complexity of developing a universally applicable verification method across various content types and generation models.

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

# A    Detailed Related Work

**Authorship Identification for Image Generation Models**

With the advancements of deep learning techniques, multiple works have suggested leveraging neural networks to seamlessly encode concealed information within images in a fully trainable manner (Zhu et al., 2018; Yang et al., 2019; Ahmadi et al., 2020; You et al., 2020). Inspired by this idea, Fernandez et al. (2022) incorporate watermarks into the latent space formulated by a self-supervised network such as DINO (Caron et al., 2021). This approach modulates the features of the image within a specific region of the latent space, ensuring that subsequent transformations applied to watermarked images preserve the integrity of the embedded information. Subsequently, watermark detection can be conducted based on this same latent space. Similarly, Fernandez et al. (2023) introduce a binary signature directly into the decoder of a diffusion model, resulting in images that contain an imperceptibly embedded binary signature. This binary signature can be accurately extracted using a pre-trained watermark extractor during verification.

Given the escalating concerns regarding the misuse of deep fakes, as highlighted in the literature (Brundage et al., 2018; Harris, 2019), several studies have proposed methodologies for attributing the origin of an image, specifically discerning between machine-generated and authentic images. This task is rendered feasible through the identification of subtle, yet machine-detectable, patterns unique to images generated by Generative Adversarial Networks (GANs), as evidenced in recent research (Marra et al., 2019; Afchar et al., 2018; Güera & Delp, 2018; Yu et al., 2019). Furthermore, the detection of deep fakes is enhanced by analyzing inconsistencies in the frequency domain or texture representation between authentic and fabricated images, as indicated in recent studies (Zhang et al., 2019; Durall et al., 2020; Liu et al., 2020).

**Authorship Identification for Natural Language Generation Models**

Content generated by text generation models is increasingly vulnerable to various forms of misuse, including the spread of misinformation and the training of surrogate models (Wallace et al., 2020; Xu et al., 2022). Consequently, a growing interest has been in protecting the authorship (or IP) of text generation models or detecting machine-generated text.

A straightforward solution is to incorporate watermarks into the generated text. However, unlike images, textual information is composed of discrete tokens, making the watermarking process for text a difficult endeavor due to the potential for inadvertent alternation that can change its semantic meaning (Katzenbeisser & Petitolas, 2000). One solution to preserve semantic integrity during watermarking involves synonym substitution (Topkara et al., 2006; Chang & Clark, 2014; He et al., 2022a). Nevertheless, the simplistic approach to synonym substitution is vulnerable to detection through statistical analyses. In response, He et al. (2022b) have proposed a conditional synonym substitution method to enhance both stealthiness and robustness of substitution-based watermarks. Moreover, Venugopal et al. (2011) adopted bit representation to encode semantically similar sentences, enabling the selection of watermarked sentences through bit manipulation.

The previously discussed methods have been centered on applying watermarks through post-editing. However, with the emergence of LLMs, there has been a significant shift towards developing watermarks tailored for LLMs to identify machine-generated text. A notable approach in this area employs biased sampling strategies to alter token distribution at each generation step, favoring tokens from specific pre-defined categories (Kirchenbauer et al., 2023a;b; Zhao et al., 2023a). Despite their innovation, these methods face vulnerabilities to "rewriting" attacks, where watermarked sentences are paraphrased either automatically or manually, thus challenging the identification of original authorship (Christ et al., 2023). To address this issue, Kuditipudi et al. (2023) propose a robust approach that maps a sequence of random numbers generated from a randomized watermark key to the outputs of a language model. This technique maintains the watermark's detectability, notwithstanding text alterations such as substitutions, insertions, or deletions.

Interest in post-hoc detection has surged as a complementary measure to watermarking. This trend is driven by the fact that developers often keep the details of their watermarking algorithms confidential to prevent them from being compromised if leaked. Depending on whether machine-generated or human-authored text samples are available, one can utilize either zero-shot or training-based detection methods. Zero-shot

detection relies on the premise that language model-generated texts inherently contain detectable markers, identifiable through analysis techniques such as perplexity (Idnay et al., 2022; Tian, 2023) or average per-token log probability (Solaiman et al., 2019a; Mitchell et al., 2023). The training-based approaches leverage features from pre-trained language models to distinguish between machine-generated and human-authored texts (Solaiman et al., 2019a; Bakhtin et al., 2019; Jawahar et al., 2020; Chen et al., 2023). However, these approaches yield a binary outcome, classifying texts as machine-generated or human-generated. Instead, our approach can determine the origin of text generated by any LLMs, moving beyond binary outcomes.

# B Experimental setup for image generation

## B.1 Embedding Watermark through Inpainting

To analyze the stability and convergence properties of inpainting models, we perform an iterative masked image infilling procedure. Given an input image $\boldsymbol{x}$ from model $\mathcal{G}$, we iteratively inpaint with mask $M$:

$$\boldsymbol{x}^{\langle k+1 \rangle} = \mathcal{G}(\boldsymbol{x}^{\langle k \rangle}, M) \tag{10}$$

Here, the mask $M$ not only guides the inpainting but also functions as the medium to embed our watermark. As we iteratively inpaint using a mask $M$, the watermark becomes more deeply embedded, serving as a distinctive signature to identify the authentic model $\mathcal{G}_a$.

### B.1.1 Convergence of Watermarked Images

The iterative masked inpainting procedure displays consistent convergence behavior across models. With a fixed binary mask covering 1/10th of the image, the distance between successive image generations decreases rapidly over the first few iterations before stabilizing. This is evidenced by the declining trend in metrics like MSE, LPIPS, and CLIP similarity as iterations increase.

The early convergence suggests the generative models are effectively reconstructing the masked regions in a coherent manner. While perfect reconstruction is infeasible over many passes, the models appear to reach reasonable stability within 5 iterations as shown in Figure 10 and 11.

Convergence to a stable equilibrium highlights latent fingerprints in the model behavior. The consistent self-reconstruction statistics form the basis for distinguishing authentic sources in the subsequent fingerprinting experiments. The watermarking convergence analysis highlights model stability and confirms that iterative inpainting effectively removes embedded watermarks without degrading image quality (see Figures in D.3.2).

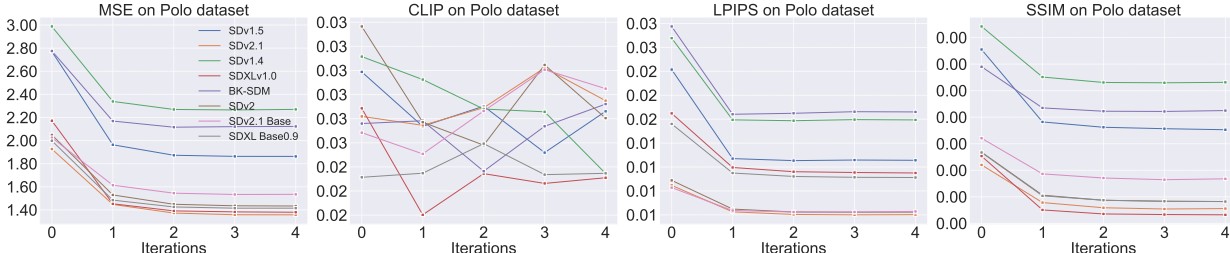

Figure 10: The convergence analysis of the distances in iteration based on various metrics on the watermarked images of 200 samples from Polo datasets.

## B.2 Enhancing and Verifying Fingerprint through Re-generation

We divide an image into non-overlapping segments and have models reconstruct masked regions to expose inherent fingerprints. Given an image $x$, we generate segment masks $M_1, ..., M_T$ covering $x$ and apply $\mathcal{G}$ to reconstruct each part:

$$\boldsymbol{y}[t] = \mathcal{G}(\boldsymbol{x}, M[t]) \tag{11}$$

By analyzing reconstructed segments $\boldsymbol{y}[t]$ and the composited image $\boldsymbol{y}$, model-specific artifacts can be quantified without additional pattern information. The non-overlapping reconstructed segments are merged to form the composite image $y$, we can obtain this by using the corresponding mask and extracting the reconstructed portion. A salient feature of our proposed re-generation paradigm is its independence from any additional information about the image partitioning pattern. The model's unique fingerprint emerges naturally during re-generation, regardless of how the image is divided or the components are merged. We experiment on 5 latest SD models - SDv 2.1, SD v2.1B, SD v2, SDXL 1.0, and SDXL0.9B using 8 segmented masks each covering 1/8th of the image thereby totaling full image coverage. Model fingerprints are identified through LPIPS, and CLIP similarities between original $x_a$ and reconstructed $y$.

### B.2.1 Convergence of Re-generated Images

Enhancing fingerprint re-generation shows consistent convergence in perceptual metrics like CLIP and LPIPS within four iterations (see Figure 12 and 13). However, traditional metrics such as MSE and SSIM lack clear convergence, suggesting inpainting effectively captures visual content but not at the pixel level. The models converge to unique stable points, revealing inherent fingerprints based on their biases and training data. This divergence is important for model attribution. Overall, re-generation effectively exposes these fingerprints while maintaining visual integrity, underscoring perceptual superiority over pixel-based metrics in evaluating generative model fingerprints.

## C Computer Vision Supplementary Experiments and Details

### C.1 Image Generation Models

We consider eight models based on the Stable Diffusion architecture (Rombach et al., 2022). These models leverage the architecture and differ primarily in their training schemes. The models are selected to span a range of architectures, training schemes, and dataset sizes. This diversity allows us to explore model-specific behaviors for attribution and stability analysis.

All models support inpainting, allowing images to be edited given a mask and image. We utilize the inpainting pipeline - StableDiffussion and StableDiffusionXL - provided by HuggingFace (von Platen et al., 2022).[7] Both SD v1.4 and SD v2 checkpoints were initialized with the weights of the SD v1.4 checkpoints and subsequently fine-tuned on "laion-aesthetics v2 5+". SDXL 1.0 employs a larger UNet backbone and a more potent text encoder (Podell et al., 2023). BKSDM is designed for efficient text-to-image synthesis, removing blocks from the U-Net and undergoing distillation pre-training on 0.22M LAION text-image pairs (Kim et al., 2023).

---

[7]https://huggingface.co/docs/diffusers/api/pipelines/stable_diffusion/inpaint

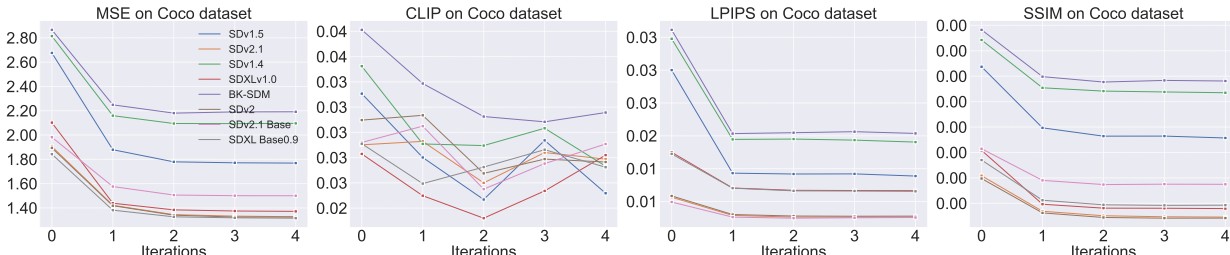

Figure 11: The convergence analysis of the distances in iterations based on various metrics on the watermarked images of 200 samples from Coco dataset

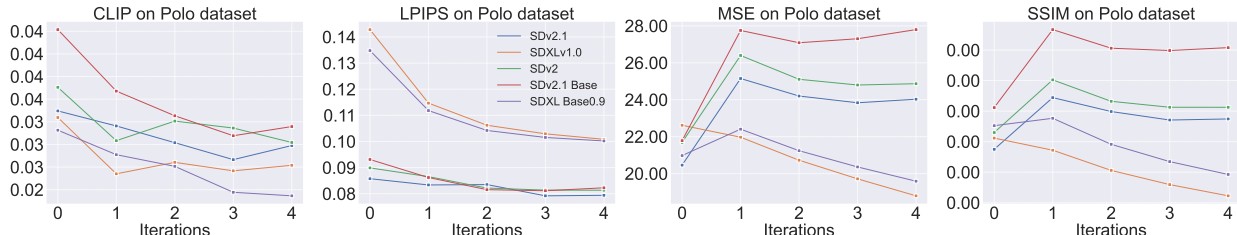

Figure 12: The convergence analysis of the distances in iteration based on various metrics on the re-generated images of 200 samples from Polo dataset.

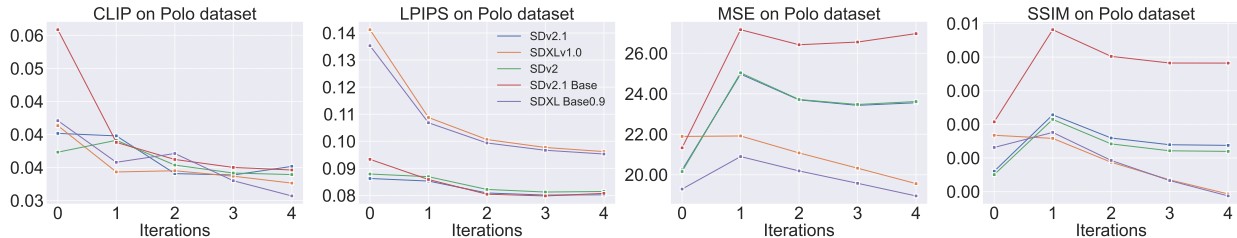

Figure 13: The convergence analysis of the distances in iteration based on various metrics on the re-generated images of 200 samples from Coco dataset.

**Image Inpainting:** Image inpainting refers to filling in missing or masked regions of an image to reconstruct the original intact image. A fixed binary mask is applied to cover certain areas of the image. The binary masks are generated as blank images filled with random white pixels. To reconstruct images, the binary mask and original image are passed to the generative model's inpainting pipeline, where the values of the pixels in the masked areas are predicted based on the unmasked context. After inpainting, the reconstructed masked regions are merged back into the re-generated image for future iteration. We utilize the inpainting pipeline[8] - StableDiffussion and StableDiffusionXL - provided by HuggingFace (von Platen et al., 2022) that enables us to regenerate a given image multiple times by masking different parts of the image.

## C.2   Distance Metrics for Images

**Contrastive Language-Image Pretraining (CLIP) Cosine Distance**   The CLIP model encodes images into high-dimensional feature representations that capture semantic content and meaning (Radford et al., 2021). With pre-trained image-text models, we can easily capture the semantic similarity of two images, which is their shared meaning and content, regardless of their visual appearance.

CLIP uses a vision transformer model as the image encoder. The output of the final transformer block can be interpreted as a semantic feature vector describing the content of the image. Images with similar content will have feature vectors close together or aligned in the embedding space.

To compare two images I1 and I2 using CLIP, we can encode them into feature vectors $f1$ and $f2$. The cosine distance between these semantic feature vectors indicates the degree of semantic alignment.

$$\text{cosine\_sim}(f_1, f_2) = \frac{f_1 \cdot f_2}{\|f_1\| \|f_2\|} \tag{12}$$

$$\text{cosine\_dist}(f_1, f_2) = 1 - \text{cosine\_sim}(f_1, f_2) \tag{13}$$

Where $\cdot$ denotes the dot product and $\| \cdot \|$ denotes the $L_2$ norm. The cosine distance ranges from 0 to 1, with 0 indicating perfectly aligned features. A lower distance between images implies more similar high-

---

[8]https://huggingface.co/docs/diffusers/api/pipelines/stable_diffusion/inpaint

Table 9: Summary of models based on the Stable Diffusion architecture for IG experiments. The **bold** models are used in our primary experiments.

| Model | Description |
|---|---|
| **Model$_1$** | Stable Diffusion 1.5: Fine-tuned from SD1.4 for 595k steps (Rombach et al., 2022) |
| **Model$_2$** | Stable Diffusion 2.1 (**SDv 2.1**): Fine-tuned from SDvTwo for 210k steps (Rombach et al., 2022) |
| **Model$_3$** | Stable Diffusion 1.4 (SD v1.4) : Initialized from SDv1.2 weights (Rombach et al., 2022) |
| **Model$_4$** | Stable Diffusion XL 1.0 (**SDXL 1.0**): Larger backbone, trained on LAION-5B (Podell et al., 2023) |
| **Model$_5$** | Block-removed Knowledge-distilled Stable Diffusion Model (Kim et al., 2023) |
| **Model$_6$** | Stable Diffusion 2 (**SD v2**) (Rombach et al., 2022) |
| **Model$_7$** | Stable Diffusion v 2.1 Base (**SD v2.1B**) (Rombach et al., 2022) |
| **Model$_8$** | Stable Diffusion XL 0.9 (**SDXL0.9B**) (Podell et al., 2023) |

level content and meaning in the images as captured by the CLIP feature embeddings. We specifically use OpenClip with a ConvNext-XXLarge encoder pretrained on laion2b dataset.

**Learned Perceptual Image Patch Similarity (LPIPS)** LPIPS metric focuses on perceptual and stylistic similarities by using a convolutional neural network pre-trained on human judgments of image patch similarities. The distance is measured between the CNN's intermediate feature representations of two images (Zhang et al., 2018).

$$d(x, x') = \sum_l \frac{1}{H_l W_l} \sum_{h,w} \left\| w_l \odot (\hat{y}_l^{hw} - \hat{y}_l'^{hw}) \right\|_2^2 \tag{14}$$

By comparing features across corresponding layers of the CNN, LPIPS can provide a fine-grained distance measuring subtle perceptual differences imperceptible in pixel space (Zhang et al., 2018). For example, changes to color scheme or artistic style that maintain semantic content will have a low CLIP distance but a higher LPIPS distance. We use the original implementation of Zhang (2023)

**Mean Squared Error (MSE)** The mean squared error (MSE) between two images $x$ and $x'$ is calculated as:

$$MSE(x, x') = \frac{1}{mn} \sum_{i=0}^{m-1} \sum_{j=0}^{n-1} [x(i,j) - x'(i,j)]^2$$

Where $x$ and $x'$ are $m \times n$ images represented as matrices of pixel intensities. The MSE measures the average of the squared intensity differences between corresponding pixels in $x$ and $x'$.

It provides a simple pixel-level similarity metric sensitive to distortions like noise, blurring, and coloring errors. However, MSE lacks perceptual relevance and is not robust to geometric/structural changes in the image.

**Structural Similarity Index (SSIM)** The structural similarity index (SSIM) (Hore & Ziou, 2010) compares corresponding $8 \times 8$ windows in the images across three terms - luminance, contrast, and structure:

$$SSIM(x, x') = \frac{(2\mu_x \mu_{x'} + c_1)(2\sigma_{xx'} + c_2)}{(\mu_x^2 + \mu_{x'}^2 + c_1)(\sigma_x^2 + \sigma_{x'}^2 + c_2)}$$

Where $\mu_x$, $\mu_{x'}$ are mean intensities, $\sigma_x^2$, $\sigma_{x'}^2$ are variances, and $\sigma_{xx'}$ is covariance for windows in $x$ and $x'$. $c_1$, $c_2$ stabilize division.

This decomposes similarity into comparative measurements of structure, luminance, and contrast. As a result, SSIM better matches human perceptual judgments compared to MSE. Values range from -1 to 1, with 1 being identical local structure.

As research rapidly improves generation quality, we expect future use cases to leverage such advanced generators. Analyzing these models is thus more indicative of real-world conditions going forward compared to earlier versions. Furthermore, the marked quality improvements in recent SD models present greater challenges for attribution and fingerprinting. Subtle inter-model differences become more difficult to quantify amidst high-fidelity outputs. Distance metrics like MSE and LPIPS are sensitive to quality, so lower baseline distortion is a more rigorous test scenario. By evaluating cutting-edge models without inpainting specialization, we aim to benchmark model fingerprinting efficacy on contemporary quality levels. Our experiments on the latest SD variants at scale also assess generalization across diverse high-fidelity generators. Successful attribution and stability analysis under these conditions will highlight the viability of our proposed techniques in real-world deployment.

### C.3 Empirical Estimation of the Lipschitz Constant $L$

For the empirical estimation of $L$, we use all pairs of images ($\boldsymbol{x}$ and $\boldsymbol{y}$) from Polo dataset, representing them as inputs and applying the re-generation function $f(\cdot)$. We measure their distances using $\mathbb{D}$, i.e. Euclidean and LPIPS. The ratio below is the transformation of Equation 8:

$$L \leq \frac{\mathbb{D}(\boldsymbol{x}, \boldsymbol{y})}{\mathbb{D}(f(\boldsymbol{x}), f(\boldsymbol{y}))} \tag{15}$$

Table 10: Lipschitz Constant Estimation

| Model | L_LPIPS Mean | L_LPIPS Std | L_Euclidean Mean | L_Euclidean Std |
|---|---|---|---|---|
| **SD v2.1** | 0.976 | 0.015 | 1.000 | 0.001 |
| **SDXL 1.0** | 0.943 | 0.21 | 1.000 | 0.004 |
| **SD v2** | 0.975 | 0.016 | 1.000 | 0.001 |
| **SD v2.1B** | 0.970 | 0.023 | 1.000 | 0.002 |
| **SDXL 0.9** | 0.946 | 0.020 | 0.999 | 0.005 |

From Table 10, we csn observe LPIPS as a distance metric consistently produced lower L values compared to Euclidean distance. This demonstrates LPIPS as a superior metric for our framework, explaining why Euclidean distance fails for verification. Additionally, more advanced models exhibited improved L values, indicating better pertubation resilience and verification capacity. Our approach focuses on leveraging inherent model properties for verification rather than enforcing universal viability. For models currently unsupported, we suggest targeting enhancements to model robustness in line with these observations.

Overall, the experiment validated LPIPS as an optimal distance function and revealed a correlation between model advancement and verifiability via intrinsic fingerprints.

### C.4 Sample Prompts and Images Reproduced

In our computer vision experiments, we sample prompts from the MS-COCO 2017 Evaluation (Lin et al., 2014) and POLOCLUB (POLO) Diffusion Dataset (Wang et al., 2022) for image generation. We present a few example prompts and images produced for different models in the section below.

### C.4.1 COCO Dataset Prompts

- A motorcycle with its brake extended standing outside.

- Off white toilet with a faucet and controls.

- A group of scooters rides down a street.

Table 11: The precision and recall of verifying the authentic models $\mathcal{G}_a$ using different contrast models $\mathcal{G}_c$ on Coco Dataset.

| $\mathcal{G}_a$ \ $\mathcal{G}_c$ | SD v2.1 | | SD v2. | | SD v2.1B | | SDXL 0.9 | | SDXL 1.0 | |
|---|---|---|---|---|---|---|---|---|---|---|
| | Precision ↑ | Recall ↑ | Precision ↑ | Recall ↑ | Precision ↑ | Recall ↑ | Precision ↑ | Recall ↑ | Precision ↑ | Recall ↑ |
| (a) $k = 1$ | | | | | | | | | | |
| SD v2.1 | - | - | 87.5 | 90.5 | 97.5 | 97.5 | 96.5 | 98.0 | 97.0 | 97.5 |
| SD v2. | 89.0 | 91.5 | - | - | 97.0 | 98.0 | 96.0 | 97.0 | 95.5 | 97.5 |
| SD v2.1B | 14.5 | 19.0 | 2.0 | 3.0 | - | - | 47.5 | 53.0 | 48.5 | 56.0 |
| SDXL 0.9 | 85.5 | 88.5 | 88.0 | 91.5 | 99.5 | 100.0 | - | - | 90.5 | 95.0 |
| SDXL 1.0 | 92.5 | 96.0 | 89.5 | 93.0 | 98.0 | 98.5 | 95.5 | 97.0 | - | - |
| (b) $k = 3$ | | | | | | | | | | |
| SD v2.1 | - | - | 73.0 | 77.0 | 97.5 | 98.5 | 94.5 | 96.5 | 95.5 | 96.5 |
| SD v2. | 72.0 | 79.0 | - | - | 97.0 | 99.0 | 95.0 | 96.0 | 94.5 | 95.5 |
| SD v2.1B | 74.0 | 79.0 | 74.5 | 79.5 | - | - | 82.0 | 86.0 | 81.5 | 85.5 |
| SDXL 0.9 | 88.5 | 90.0 | 90.0 | 91.5 | 98.0 | 98.5 | - | - | 86.5 | 88.0 |
| SDXL 1.0 | 91.5 | 94.5 | 92.5 | 95.0 | 100.0 | 100.0 | 89.0 | 91.0 | - | - |
| (c) $k = 5$ | | | | | | | | | | |
| SD v2.1 | - | - | 66.0 | 73.0 | 99.5 | 99.5 | 94.5 | 97.0 | 93.0 | 96.0 |
| SD v2. | 72.5 | 81.5 | - | - | 96.0 | 96.5 | 91.0 | 92.5 | 94.0 | 95.0 |
| SD v2.1B | 78.5 | 83.5 | 77.0 | 82.5 | - | - | 82.5 | 87.0 | 85.5 | 89.5 |
| SDXL 0.9 | 97.0 | 98.0 | 97.5 | 98.0 | 100.0 | 100.0 | - | - | 92.0 | 95.0 |
| SDXL 1.0 | 93.0 | 96.0 | 92.5 | 93.0 | 99.0 | 99.5 | 88.0 | 90.5 | - | - |

### C.4.2 Polo Dataset Prompts

- A renaissance portrait of Dwayne Johnson, art in the style of Rembrandt!! Intricate. Ultra detailed, oil on canvas, wet-on-wet technique, pay attention to facial details, highly realistic, cinematic lightning, intricate textures, illusionistic detail.

- Epic 3D, become legend shiji! GPU mecha controlled by telepathic hackers, made of liquid, bubbles, crystals, and mangroves, Houdini SideFX, perfect render, ArtStation trending, by Jeremy Mann, Tsutomu Nihei and Ilya Kuvshinov.

- An airbrush painting of a cyber war machine scene in area 5 1 by Destiny Womack, Gregoire Boonzaier, Harrison Fisher, Richard Dadd.

### C.5 Density Distribution of a one-step re-generation

The one-step re-generation density distributions reveal distinct model-specific characteristics, enabling discrimination as seen in Figure 14 and 15, Most models exhibit distinction, with each distribution showing unique traits.

A notable exception in this behavior is observed for the SD v2.1B (see Figures 14 and 15). which initially demonstrates less discrimination. However, over extended iterations, SD v2.1B shows marked improvement, highlighting the capacity for iterative refinement. By the 5th iteration, there is a noticiable improvement in the discriminative nature of its one-step re-generation. This improvement is crucial as it highlights the model's capacity to refine and enhance its re-generative characteristics over time.

While LPIPS is not immediately effective in pinpointing the authentic model at the very first step, it still offers a powerful mechanism to distinguish between models. LPIPS is effective at differentiating between families of models, such as the Stable Diffusion models and the Stable Diffusion XL models, as visualized in Figure 16 for the Polo dataset and Figure 17 for the Coco dataset. The lack of effectiveness of LPIPS in identifying the authentic model is a primary reason why it was not chosen for verification.

Further insights into the models' discriminative capabilities can be derived from Table 11 and 12, while SDv2.1B starts with lower accuracy in distinguishing itself, a significant improvement in accuracy is seen across iterations. The initially anomalous behavior transitions into more discriminating re-generation.

Table 12: The precision and recall of verifying the authentic models $\mathcal{G}_a$ using different contrast models $\mathcal{G}_c$ on Polo Dataset.

| $\mathcal{G}_c$ / $\mathcal{G}_a$ | SD v2.1 | | SD v2. | | SD v2.1B | | SDXL 0.9 | | SDXL 1.0 | |
|---|---|---|---|---|---|---|---|---|---|---|
| | Precision ↑ | Recall ↑ | Precision ↑ | Recall ↑ | Precision ↑ | Recall ↑ | Precision ↑ | Recall ↑ | Precision ↑ | Recall ↑ |
| (a) $k=1$ | | | | | | | | | | |
| SD v2.1 | - | - | 76.0 | 78.5 | 97.5 | 99.0 | 93.5 | 97.0 | 94.0 | 95.0 |
| SD v2. | 79.0 | 84.0 | - | - | 98.0 | 98.5 | 92.0 | 95.5 | 94.0 | 95.5 |
| SD v2.1B | 40.5 | 51.0 | 39.0 | 46.0 | - | - | 72.5 | 80.0 | 73.5 | 80.5 |
| SDXL 0.9 | 86.0 | 88.5 | 85.0 | 89.5 | 99.5 | 99.5 | - | - | 91.5 | 95.5 |
| SDXL 1.0 | 89.0 | 92.0 | 89.5 | 93.0 | 98.0 | 99.0 | 91.5 | 95.5 | - | - |
| (b) $k=3$ | | | | | | | | | | |
| SD v2.1 | - | - | 77.0 | 83.0 | 99.5 | 100.0 | 95.5 | 98.0 | 96.5 | 97.5 |
| SD v2. | 75.5 | 82.0 | - | - | 98.0 | 98.0 | 96.5 | 98.0 | 97.5 | 97.5 |
| SD v2.1B | 70.0 | 76.5 | 71.5 | 78.5 | - | - | 84.0 | 90.5 | 84.5 | 89.5 |
| SDXL 0.9 | 92.5 | 95.0 | 91.0 | 94.0 | 100.0 | 100.0 | - | - | 91.0 | 94.0 |
| SDXL 1.0 | 90.0 | 92.5 | 89.5 | 91.5 | 100.0 | 100.0 | 90.0 | 93.0 | - | - |
| (c) $k=5$ | | | | | | | | | | |
| SD v2.1 | - | - | 80.0 | 86.0 | 99.5 | 99.5 | 98.5 | 99.5 | 99.0 | 100.0 |
| SD v2. | 77.5 | 81.5 | - | - | 99.0 | 100.0 | 96.5 | 97.0 | 95.5 | 97.5 |
| SD v2.1B | 81.5 | 83.5 | 82.5 | 86.5 | - | - | 88.5 | 92.0 | 89.5 | 91.0 |
| SDXL 0.9 | 97.5 | 97.5 | 96.5 | 98.0 | 99.5 | 99.5 | - | - | 92.0 | 93.5 |
| SDXL 1.0 | 95.5 | 96.0 | 92.5 | 95.5 | 100.0 | 0.0 | 94.5 | 96.0 | - | - |

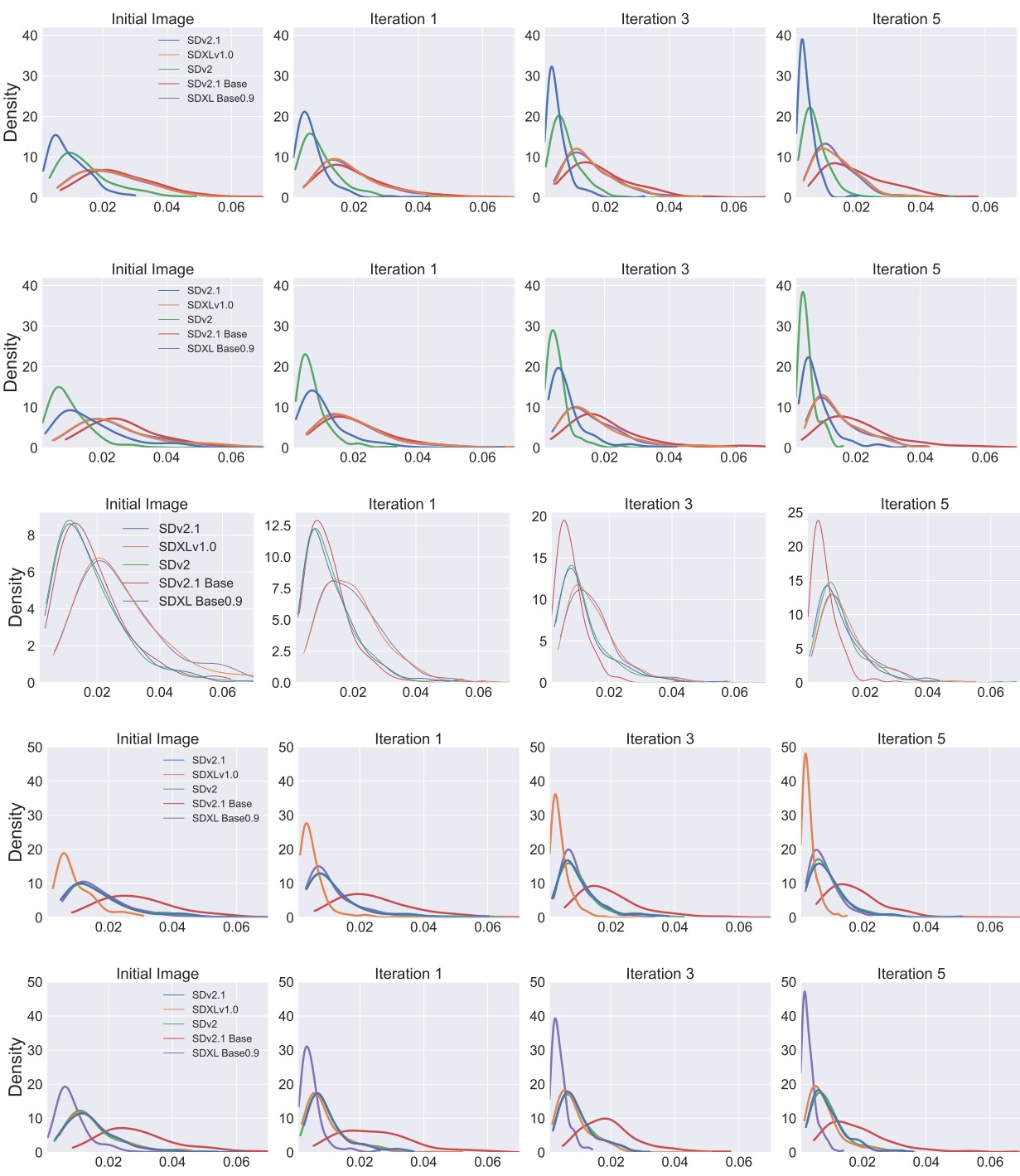

Figure 14: One Step Re-generation for various authentic models on the Polo Dataset using the CLIP metric at different iterations. The authentic models from top to bottom are: 1) SDv 2.1, 2) SD v2, 3) SD v2.1B,4) SDXL 1.0, 5) SDXL0.9B.

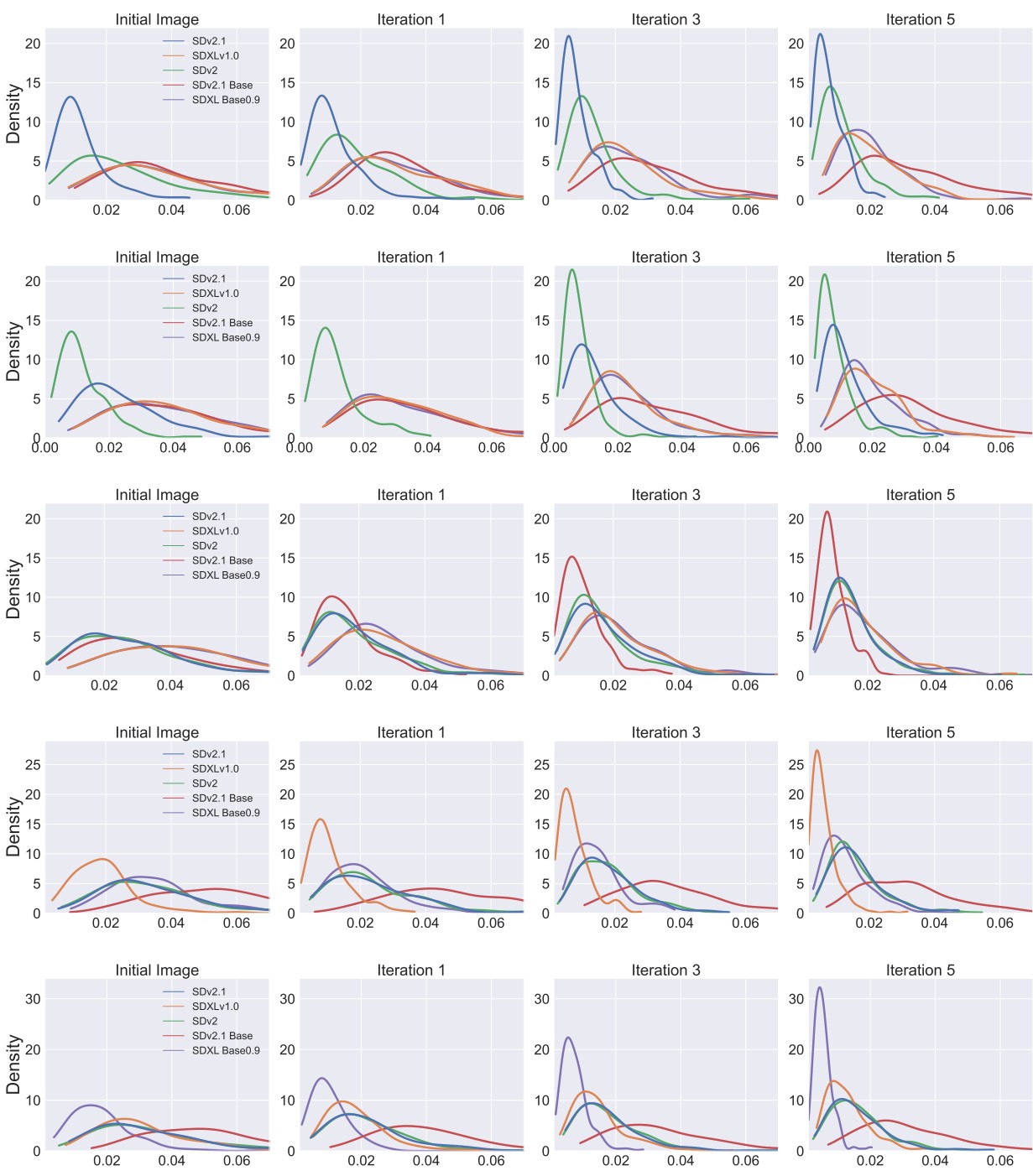

Figure 15: One Step Re-generation for various Authentic Models on the Coco Dataset using the CLIP metric at different iterations. The authentic models from top to bottom are: 1) SDv 2.1, 2) SD v2, 3) SD v2.1B, 4) SDXL 1.0, 5) SDXL0.9B.

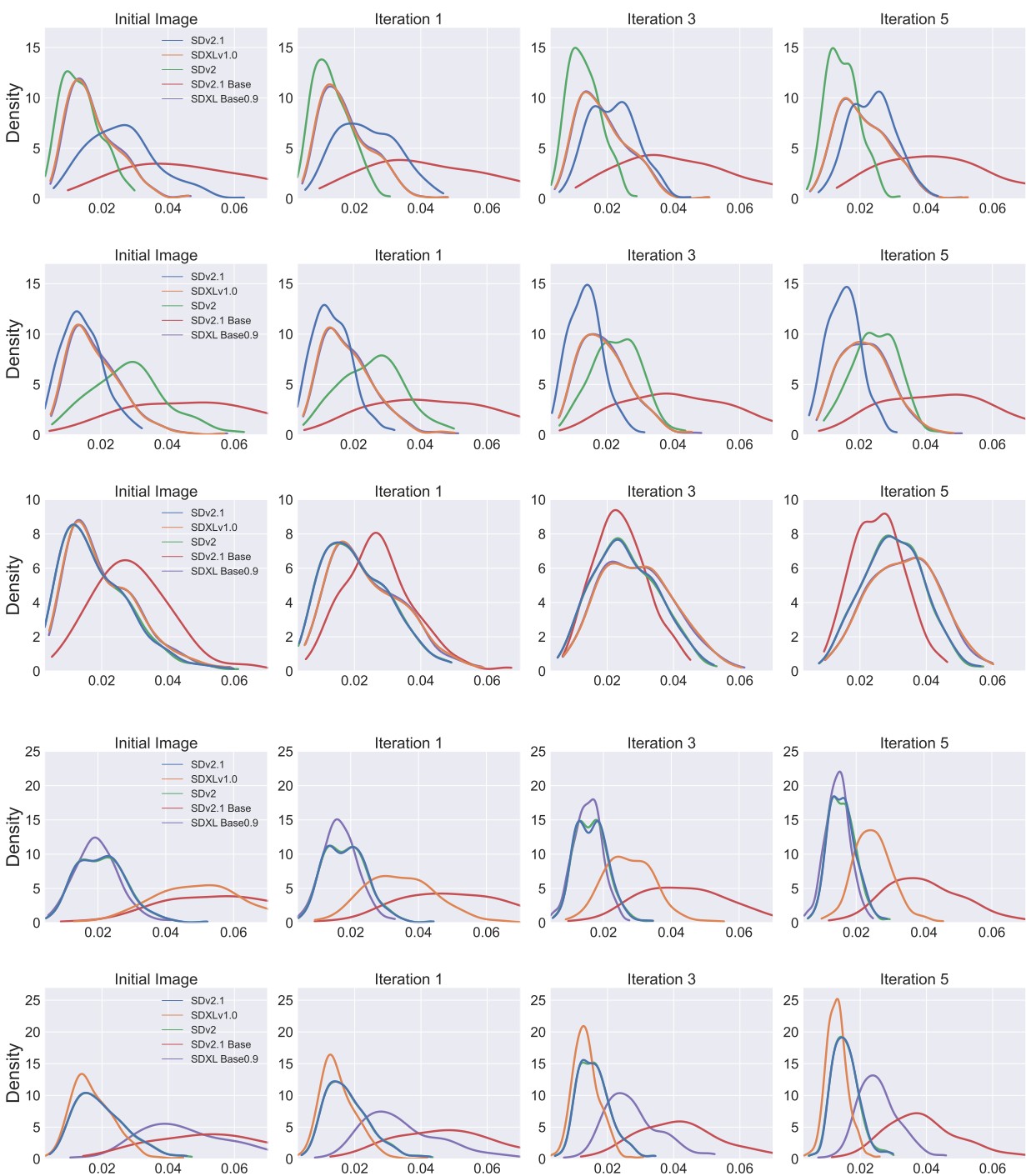

Figure 16: One Step Re-generation for various Authentic Models on the Polo Dataset using the LPIPS metric at different iterations. The authentic models from top to bottom are: 1) SDv 2.1, 2) SD v2, 3) SD v2.1B, 4) SDXL 1.0, 5) SDXL0.9B.

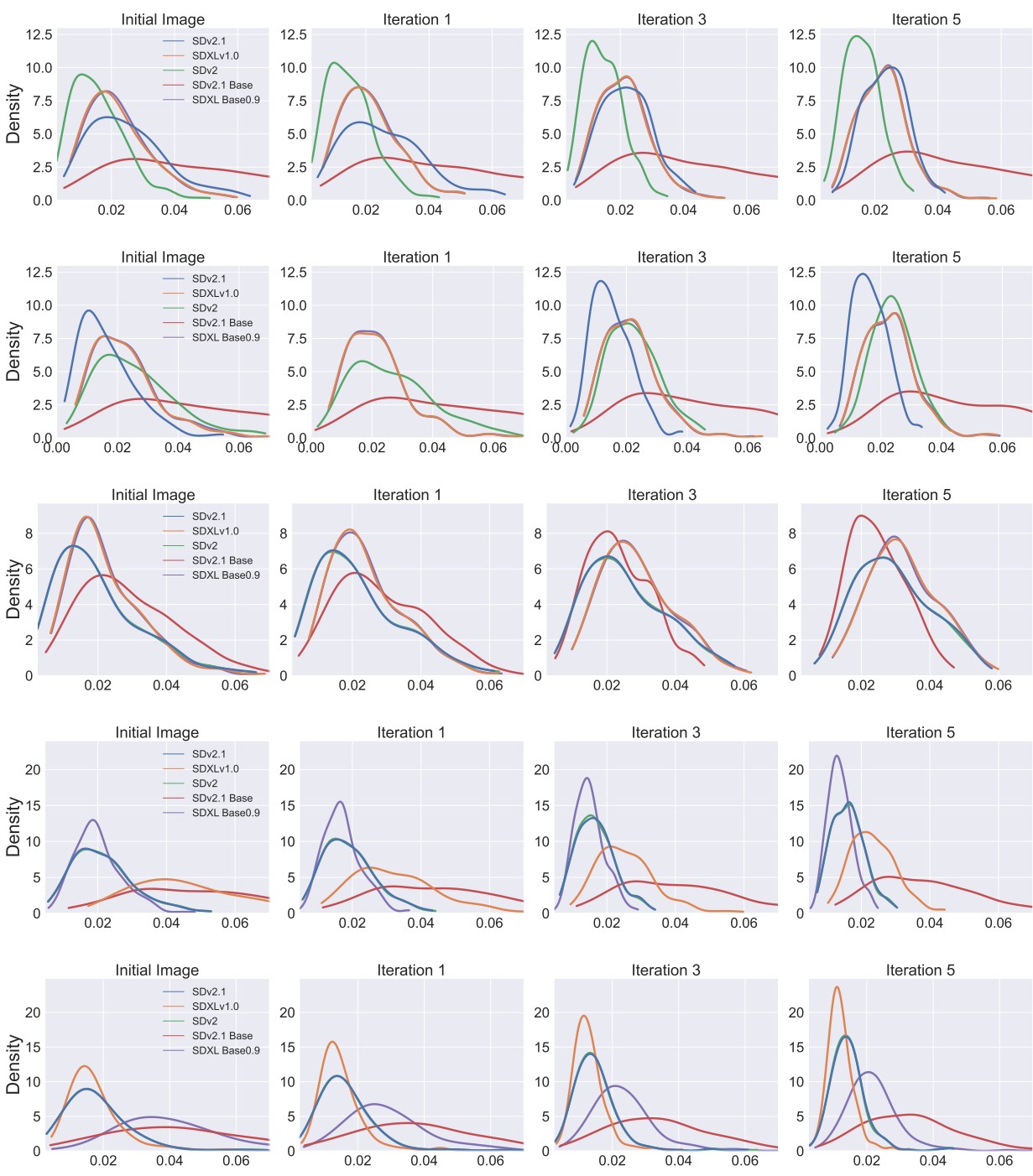

Figure 17: One Step Re-generation for various Authentic Models on the Coco Dataset using the LPIPS metric at different iterations. The authentic models from top to bottom are: 1) SDv 2.1, 2) SD v2, 3) SD v2.1B, 4) SDXL 1.0, 5) SDXL0.9B.

# D    Supplementary experiments for text generation models

## D.1    Density distribution of a one-step re-generation

In Figure 18, we illustrate the density distribution of one-step re-generation for four text generation models, using the first iteration from the authentic models as input. Excluding Cohere, the density distributions of the authentic models are discernible from those of the contrast models across BLEU, ROUGE-L, and BERTScore metrics.

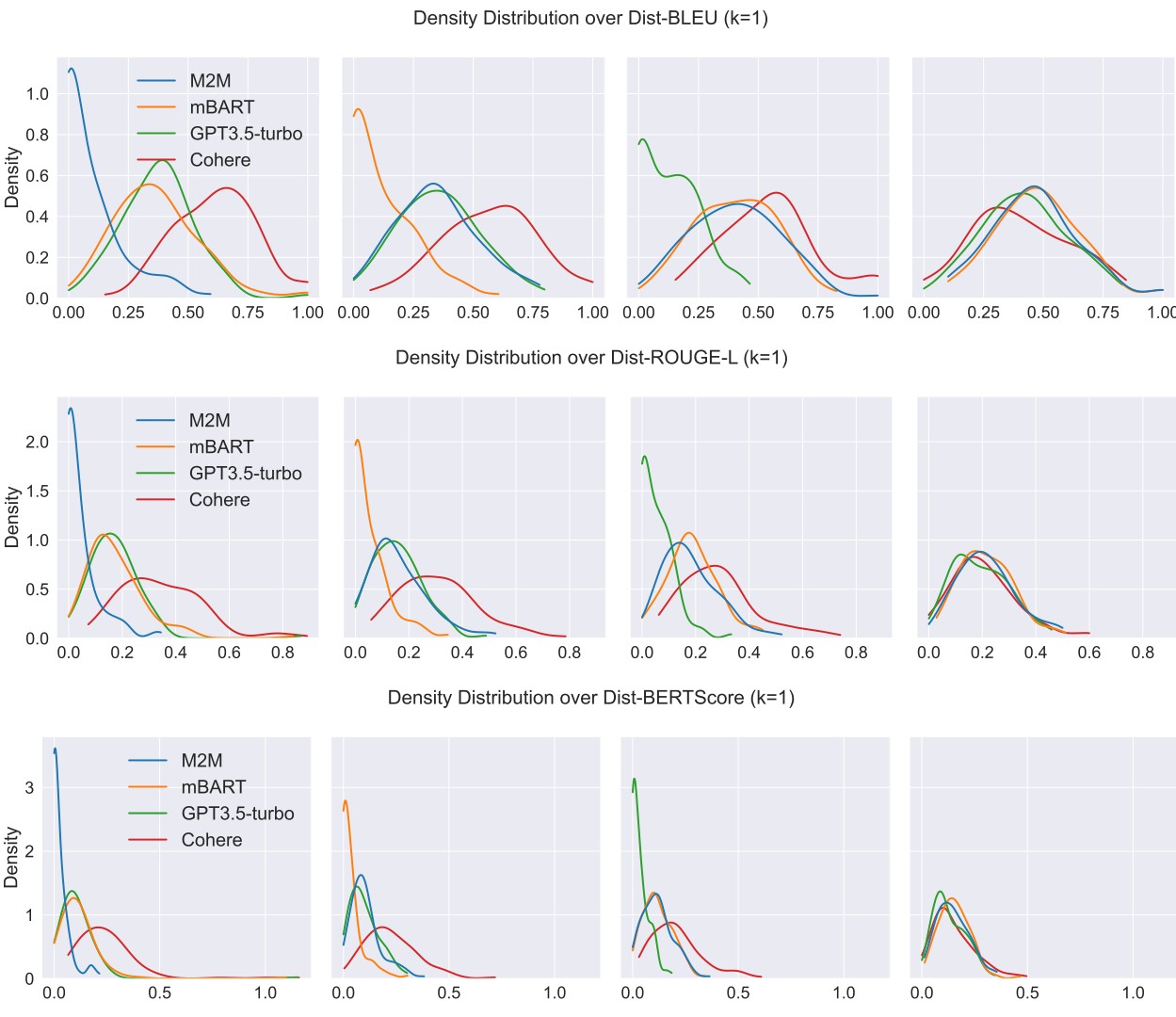

Figure 18: Density distribution of one-step re-generation among four text generation models, where the input to the one-step re-generation is the first iteration from the authentic models. The authentic models from left to right are: 1) M2M, 2) mBART, 3) GPT3.5-turbo, and 4) Cohere.

Figure 19 illustrates the density distribution when the input is the $k$th iteration from the authentic models. As $k$ increases, the distinction between the authentic model's distribution and that of the contrast models becomes more pronounced.

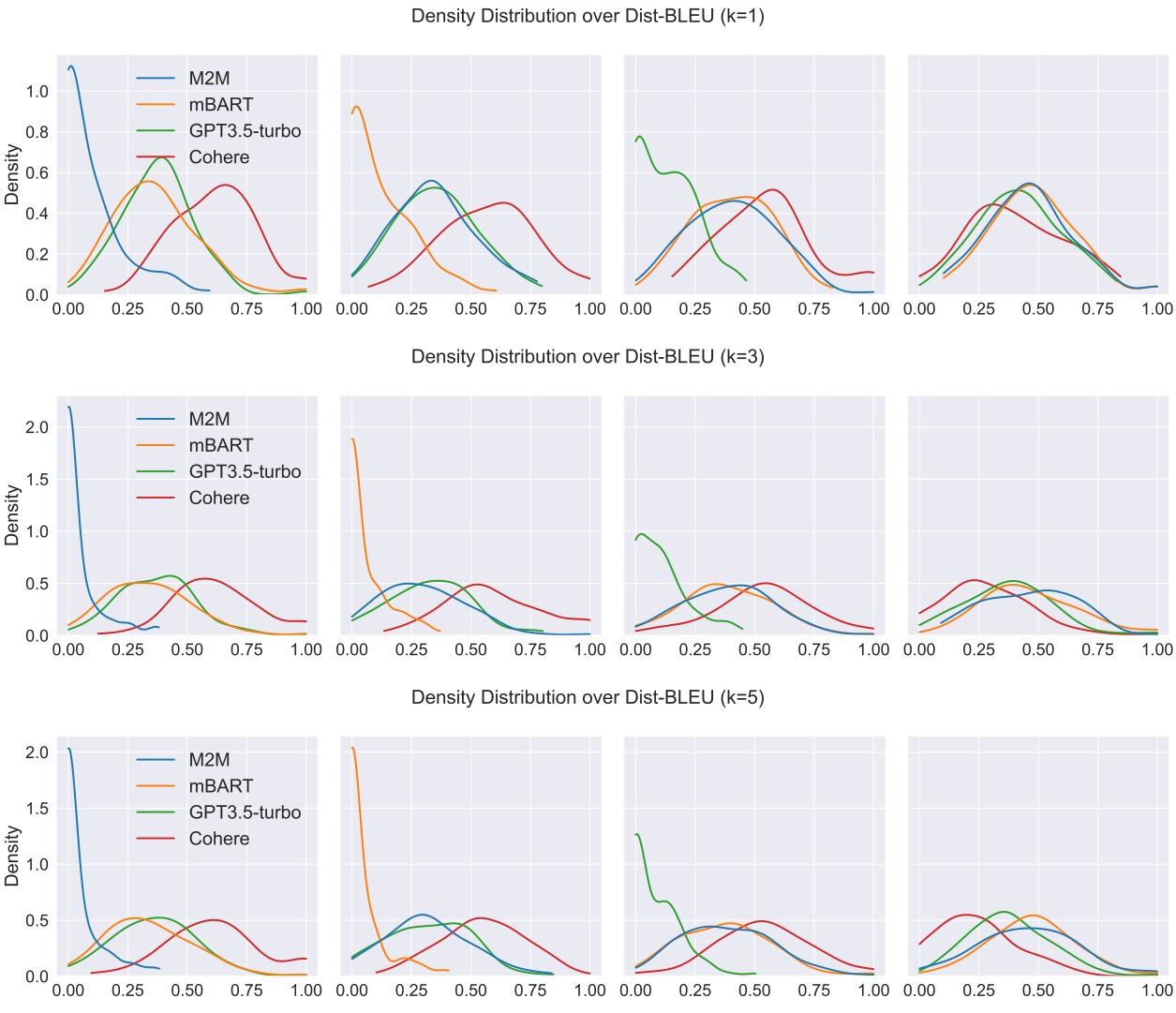

Figure 19: Density distribution of one-step re-generation among four text generation models, where the input to the one-step re-generation is the $k$th iteration from the authentic models. The authentic models from left to right are: 1) M2M, 2) mBART, 3) GPT3.5-turbo, and 4) Cohere.

### D.2 Experiments for GPT4

In this section, we analyze the characteristics of GPT4. We first present the density distribution when the input is the $k$th iteration from the authentic models in Figure 20. Similarly, as $k$ increases, the difference between the authentic model's distribution and the contrast models intensifies. Nonetheless, distinguishing between GPT3.5-turbo and GPT4 proves difficult, particularly when GPT3.5-turbo serves as the authentic model, even for larger values of $k$. Detailed examination reveals that GPT4's one-step re-generation bears resemblance to the $k$-th iteration of GPT3.5-turbo. This might stem from GPT3.5-turbo and GPT4 originating from the same institution, suggesting potential similarities in architecture and pre-training data. Thus, we contend that models stemming from the same institution inherently share identical intellectual property, thus obviating the possibility of intellectual property conflicts. Interestingly, when GPT4 is the authentic model, our methodology can differentiate it from GPT3.5-turbo, attributed to the marked difference between GPT3.5-turbo's one-step re-generation and GPT4's $k$-th iteration. This distinction, we surmise, is due to GPT4's superior advancement over GPT3.5-turbo.

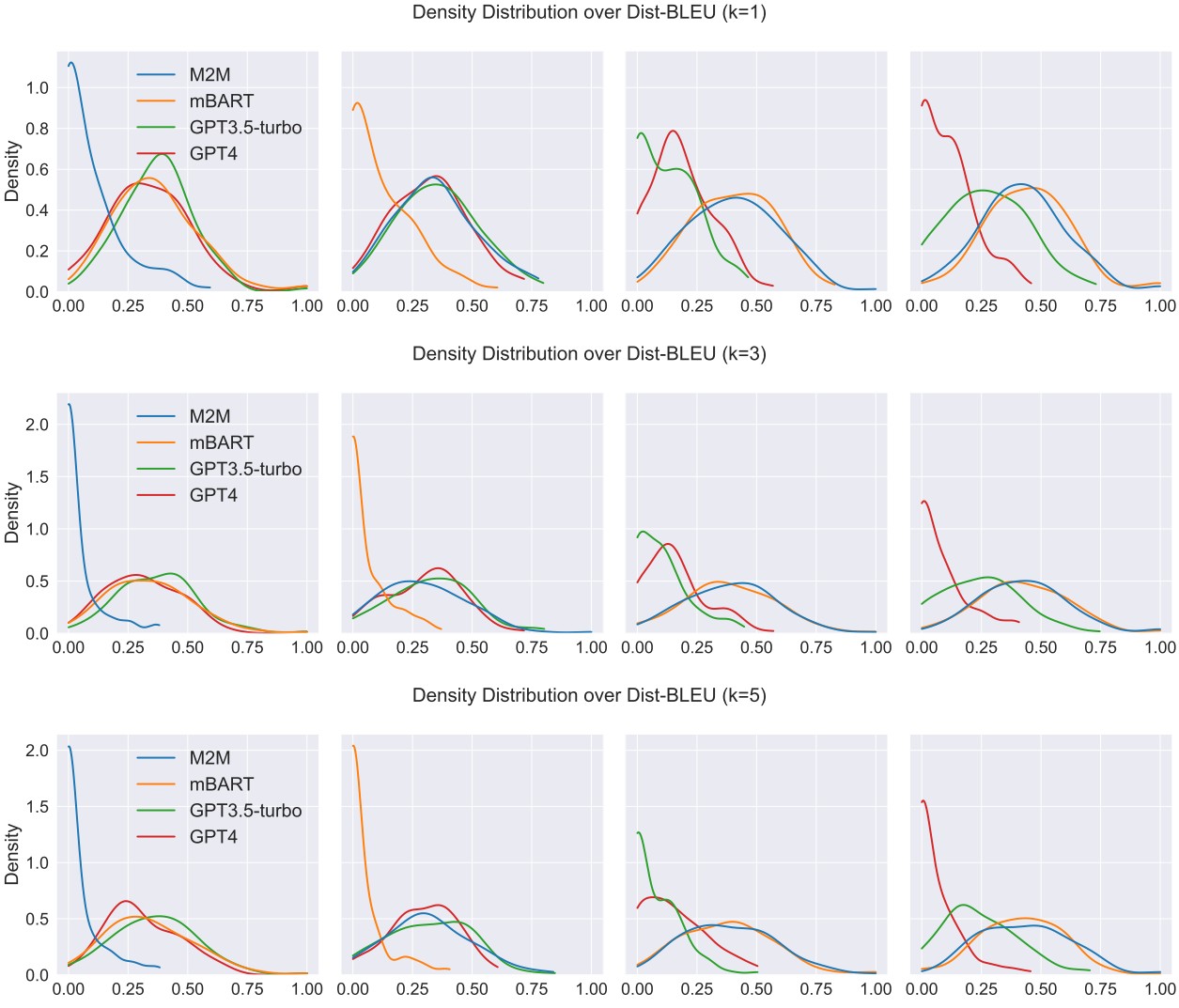

Figure 20: Density distribution of one-step re-generation among four text generation models, where the input to the one-step re-generation is the $k$th iteration from the authentic models. The authentic models from left to right are: 1) M2M, 2) mBART, 3) GPT3.5-turbo, and 4) GPT4.

In evaluating the verification of authentic models, our methodology, as depicted in Table 13, consistently confirms the authorship of all models with an accuracy exceeding 85%. Furthermore, the misclassification rate remains below 10%.

### D.3 Quality of Re-generated Sentences and Images

#### D.3.1 Text Re-generations

In this section, we examine the quality of re-generated sentences. As illustrated in Figure 21, the re-generated sentences display high quality across most evaluation metrics, except for Cohere. The slower convergence observed for Cohere can be attributed to its subpar re-generation quality, which significantly deviates from that of other models. For further illustration, refer to the samples and their re-generations in Table 14-17.

Table 13: The precision and recall of verifying the authentic models ($\mathcal{G}_a$) using different contrast models ($\mathcal{G}_c$).

| $\mathcal{G}_c$ $\mathcal{G}_a$ | M2M | | mBART | | GPT3.5-turbo | | GPT4 | |
|---|---|---|---|---|---|---|---|---|
| | Precision ↑ | Recall ↑ | Precision ↑ | Recall ↑ | Precision ↑ | Recall ↑ | Precision ↑ | Recall ↑ |
| (a) $k = 1$ | | | | | | | | |
| M2M | - | - | 94.0 | 98.0 | 93.0 | 95.0 | 87.0 | 92.0 |
| mBART | 85.0 | 89.0 | - | - | 89.0 | 92.0 | 80.0 | 85.0 |
| GPT3.5-turbo | 87.0 | 92.0 | 90.0 | 92.0 | - | - | 51.0 | 69.0 |
| GPT4 | 92.0 | 98.0 | 94.0 | 98.0 | 77.0 | 90.0 | - | - |
| (b) $k = 3$ | | | | | | | | |
| M2M | - | - | 94.0 | 99.0 | 95.0 | 97.0 | 94.0 | 95.0 |
| mBART | 85.0 | 93.0 | - | - | 90.0 | 96.0 | 89.0 | 95.0 |
| GPT3.5-turbo | 89.0 | 92.0 | 93.0 | 95.0 | - | - | 51.0 | 72.0 |
| GPT4 | 96.0 | 97.0 | 92.0 | 94.0 | 79.0 | 91.0 | - | - |
| (c) $k = 5$ | | | | | | | | |
| M2M | - | - | 94.0 | 98.0 | 95.0 | 99.0 | 94.0 | 96.0 |
| mBART | 91.0 | 96.0 | - | - | 88.0 | 94.0 | 89.0 | 94.0 |
| GPT3.5-turbo | 90.0 | 94.0 | 94.0 | 98.0 | - | - | 55.0 | 83.0 |
| GPT4 | 97.0 | 98.0 | 94.0 | 97.0 | 83.0 | 95.0 | - | - |

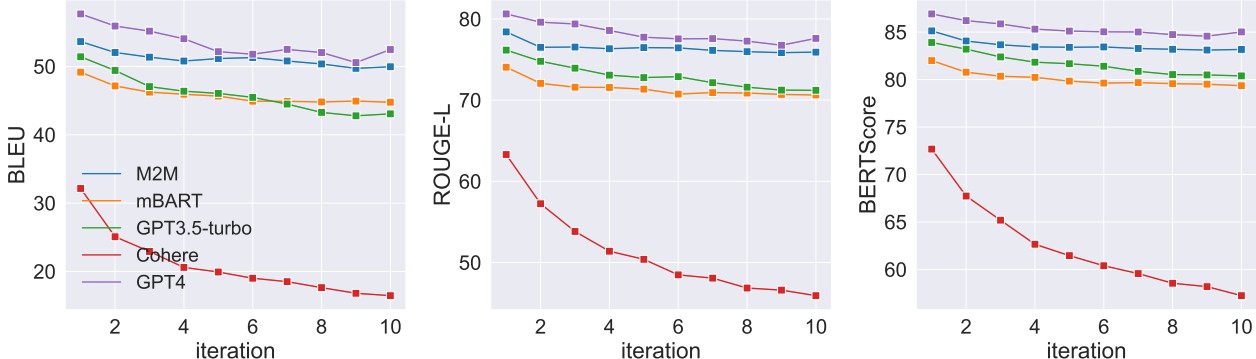

Figure 21: The quality between the original inputs and $k$th re-generation using BLEU, ROULGE-L and BERTScore.

### D.3.2 Image Re-generations

An integral part of our evaluation process involves understanding the reproducibility of the images generated by different models across iterations. To this end, we present image re-generations for various authentic models in Figure 22-26. These re-generations provide insights into the consistency and stability of each model when tasked with reproducing the same visual content over multiple iterations.

Table 14: Three original samples and their corresponding $k$-iteration from M2M.

| Sample 1: |
|---|

**Original input**: Working in groups of 2 or 4 (even numbers may work better later on), suggest a heuristic to evaluate the utility of a given board configuration.
**Iteration 1**: Working in groups of 2 or 4 (even numbers can work better later), suggests a heurist to evaluate the usefulness of a given table configuration.
**Iteration 2**: Working in groups of 2 or 4 (even numbers can work better later), suggests a heurist to evaluate the usefulness of a given table configuration.
**Iteration 3**: Working in groups of 2 or 4 (even numbers can work better later), suggests a heurist to evaluate the usefulness of a given table configuration.
**Iteration 4**: Working in groups of 2 or 4 (even numbers can work better later), suggests a heurist to evaluate the usefulness of a given table configuration.
**Iteration 5**: Working in groups of 2 or 4 (even numbers can work better later), suggests a heurist to evaluate the usefulness of a given table configuration.

| Sample 2: |
|---|

**Original input**: Therefore, to improve the ethical coping ability of Chinese NLP techniques and their application effectiveness in the field of ethics, the Chinese ethics knowledge base and KEPTMs for ethics domain are researched
**Iteration 1**: Therefore, in order to improve the ethical treatment capacity of Chinese NLP techniques and their effectiveness of application in the field of ethics, the knowledge base of Chinese ethics and the KEPTMs for the field of ethics are studied.
**Iteration 2**: Therefore, in order to improve the ethical treatment capacity of Chinese NLP techniques and their effectiveness of application in the field of ethics, the knowledge base of Chinese ethics and KEPTMs for the field of ethics are studied.
**Iteration 3**: Therefore, in order to improve the ethical treatment capacity of Chinese NLP techniques and their effectiveness of application in the field of ethics, the knowledge base of Chinese ethics and KEPTMs for the field of ethics are studied.
**Iteration 4**: Therefore, in order to improve the ethical treatment capacity of Chinese NLP techniques and their effectiveness of application in the field of ethics, the knowledge base of Chinese ethics and KEPTMs for the field of ethics are studied.
**Iteration 5**: Therefore, in order to improve the ethical treatment capacity of Chinese NLP techniques and their effectiveness of application in the field of ethics, the knowledge base of Chinese ethics and KEPTMs for the field of ethics are studied.

| Sample 3: |
|---|

**Original input**: I kindly request a modification to the due date for my assignment. According to the previous thread, I was informed that I would receive an extension of 33 hours for this assignment. However, as of now, my Moodle account still indicates the original due date
**Iteration 1**: I slowly ask for a change to the appropriate date for my task. According to the previous topic, I was informed that I would receive an extension of 33 hours for this task. However, from now on, my Moodle account still indicates the appropriate original date.
**Iteration 2**: I slowly ask for a change to the appropriate date for my task. According to the previous subject, I was informed that I would receive a 33-hour extension for this task. However, from now on, my Moodle account still indicates the appropriate original date.
**Iteration 3**: I slowly ask for a change to the appropriate date for my task. According to the previous topic, I was informed that I would receive a 33-hour extension for this task. However, from now on, my Moodle account still indicates the appropriate original date.
**Iteration 4**: I slowly ask for a change to the appropriate date for my task. According to the previous topic, I was informed that I would receive a 33-hour extension for this task. However, from now on, my Moodle account still indicates the appropriate original date.
**Iteration 5**: I slowly ask for a change to the appropriate date for my task. According to the previous topic, I was informed that I would receive an extension of 33 hours for this task. However, from now on, my Moodle account still indicates the appropriate original date.

Table 15: Three original samples and their corresponding $k$-iteration from mBART.

| Sample 1: |
| --- |

**Original input**: Working in groups of 2 or 4 (even numbers may work better later on), suggest a heuristic to evaluate the utility of a given board configuration.
**Iteration 1**: If you work in groups of 2 or 4 (even numbers can work better later), suggest a mouse kit to evaluate the usefulness of a given configuration of the table.
**Iteration 2**: If you work in groups of 2 or 4 (even numbers can work better later), suggest a mouse kit to evaluate the usefulness of a particular table configuration.
**Iteration 3**: If you work in groups of 2 or 4 (even numbers can work better later), suggest a mouse kit to evaluate the usefulness of a particular table configuration.
**Iteration 4**: If you work in groups of 2 or 4 (even numbers can work better later), suggest a mouse kit to evaluate the usefulness of a given table configuration.
**Iteration 5**: If you work in groups of 2 or 4 (even numbers can work better later), suggest a mouse kit to evaluate the usefulness of a given table configuration.

| Sample 2: |
| --- |

**Original input**: Therefore, to improve the ethical coping ability of Chinese NLP techniques and their application effectiveness in the field of ethics, the Chinese ethics knowledge base and KEPTMs for ethics domain are researched
**Iteration 1**: For this reason, to improve the ethical adaptation capacity of Chinese NLP techniques and their effective application in the ethics field, the Chinese Ethics Knowledge Base and the KEPTMs for the ethics field are being studied.
**Iteration 2**: Therefore, to improve the ethical adaptation capacity of Chinese NLP techniques and their effective application in the field of ethics, the Chinese ethics knowledge base and the ECEs for the field of ethics are being studied.
**Iteration 3**: Consequently, in order to improve the ethical adaptation capacity of Chinese PLN techniques and their effective application in the field of ethics, the Chinese ethics knowledge base and ECE for the field of ethics are being studied.
**Iteration 4**: Consequently, in order to improve the ethical adaptation capacity of Chinese PLN techniques and their effective application in the field of ethics, the Chinese ethics knowledge base and UNECE for the field of ethics are being studied.
**Iteration 5**: Consequently, in order to improve the ethical adaptation capacity of Chinese PLN techniques and their effective application in the field of ethics, the Chinese ethics knowledge base and UNECE for the field of ethics are being studied.

| Sample 3: |
| --- |

**Original input**: I kindly request a modification to the due date for my assignment. According to the previous thread, I was informed that I would receive an extension of 33 hours for this assignment. However, as of now, my Moodle account still indicates the original due date.
**Iteration 1**: I am requesting a change to the due date of my assignment, and according to the previous thread, I have been informed that I will receive a 33-hour extension for this assignment, but at the moment my Moodle account still indicates the original due date.
**Iteration 2**: I am requesting a change to my assignment due date and, according to the previous thread, I have been informed that I have received a 33-hour extension for this assignment, but at the moment, my Moodle account still indicates the original due date.
**Iteration 3**: I am requesting a change to my assignment due date and, according to the previous thread, I have been informed that I have received a 33-hour extension for this assignment, but for the time being, my Moodle account still indicates the original due date.
**Iteration 4**: I am requesting a change to the due date of my assignment and, according to the previous thread, I have been informed that I have received a 33-hour extension for this assignment, but for the time being, my Moodle account still indicates the original due date.
**Iteration 5**: I am requesting a change to the due date of my assignment and, according to the previous thread, I was informed that I have received a 33-hour extension for this assignment, but for the time being, my Moodle account still indicates the original due date.

Table 16: Three original samples and their corresponding $k$-iteration from GPT3.5-turbo.

---

Sample 1:

**Original input**: Working in groups of 2 or 4 (even numbers may work better later on), suggest a heuristic to evaluate the utility of a given board configuration.
**Iteration 1**: Working in groups of 2 or 4 (even numbers might work better later on), propose a heuristic to evaluate the usefulness of a given board configuration.
**Iteration 2**: Working in groups of 2 or 4 (even numbers might work better in the future), propose a heuristic to evaluate the usefulness of a given board configuration.
**Iteration 3**: Working in groups of 2 or 4 (even numbers might work better in the future), propose a heuristic to evaluate the utility of a given board configuration.
**Iteration 4**: Working in groups of 2 or 4 (even numbers might work better in the future), propose a heuristic to evaluate the usefulness of a given board configuration.
**Iteration 5**: Working in groups of 2 or 4 (even numbers might work better in the future), propose a heuristic to evaluate the usefulness of a given board configuration.

---

Sample 2:

**Original input**: Therefore, to improve the ethical coping ability of Chinese NLP techniques and their application effectiveness in the field of ethics, the Chinese ethics knowledge base and KEPTMs for ethics domain are researched
**Iteration 1**: Therefore, in order to improve the ethical adaptability of Chinese NLP techniques and their application efficiency in the field of ethics, the Chinese ethics knowledge base and KEPTMs for the field of ethics are studied.
**Iteration 2**: Therefore, in order to improve the ethical adaptability of Chinese natural language processing techniques and their effectiveness in the field of ethics, Chinese ethical knowledge base and KEPTMs for the field of ethics are studied.
**Iteration 3**: Therefore, in order to improve the ethical adaptability of Chinese natural language processing techniques and their effectiveness in the field of ethics, Chinese ethical knowledge base and KEPTMs for the field of ethics are studied.
**Iteration 4**: Therefore, in order to improve the ethical adaptability of Chinese natural language processing techniques and their effectiveness in the field of ethics, Chinese ethical knowledge base and KEPTMs for the field of ethics are being studied.
**Iteration 5**: Therefore, in order to improve the ethical adaptability of Chinese natural language processing techniques and their effectiveness in the field of ethics, Chinese ethical knowledge base and KEPTMs for the field of ethics are being studied.

---

Sample 3:

**Original input**: I kindly request a modification to the due date for my assignment. According to the previous thread, I was informed that I would receive an extension of 33 hours for this assignment. However, as of now, my Moodle account still indicates the original due date.
**Iteration 1**: I politely request a modification of the due date for my assignment. According to the previous thread, I was informed that I would be granted a 33-hour extension for this assignment. However, at the moment, my Moodle account still shows the original due date.
**Iteration 2**: I politely request a modification of the deadline for my assignment. According to the previous thread, I was informed that I would be granted an additional 33 hours for this assignment. However, at the moment, my Moodle account still displays the original deadline.
**Iteration 3**: I politely request a change in the deadline for my assignment. According to the previous thread, I was informed that I would have an additional 33 hours for this assignment. However, currently, my Moodle account still displays the original deadline.
**Iteration 4**: I politely request a change in deadline for my assignment. According to the previous thread, I was informed that I would have an additional 33 hours for this assignment. However, currently, my Moodle account still displays the original deadline.
**Iteration 5**: I kindly request a change of deadline for my assignment. According to the previous thread, I was informed that I would have an additional 33 hours for this assignment. However, currently, my Moodle account still displays the original deadline.

---

Table 17: Three original samples and their corresponding $k$-iteration from Cohere.

| Sample 1: |
|---|
| **Original input**: Working in groups of 2 or 4 (even numbers may work better later on), suggest a heuristic to evaluate the utility of a given board configuration. 
 **Iteration 1**: Working in groups of 2 or 4 (odd numbers may work better later), suggests a heuristic to evaluate the utility of a seating chart configuration. 
 **Iteration 2**: Working in groups of 2 or 4 (odd numbers may work better later), suggests a heuristic for evaluating the utility of a configuration of the seating plan. 
 **Iteration 3**: Working in groups of 2 or 4 (odd numbers can work better later), suggests an algorithm for evaluating the utility of a assistance plan configuration. 
 **Iteration 4**: Working in groups of 2 or 4 (odd numbers can work better later), suggests an algorithm to evaluate the utility of a plan of assistance configuration. 
 **Iteration 5**: Working in groups of 2 or 4 (odd numbers may work better later), suggests an algorithm to evaluate the utility of a configuration of assistance plans. |
| Sample 2: |
| **Original input**: Therefore, to improve the ethical coping ability of Chinese NLP techniques and their application effectiveness in the field of ethics, the Chinese ethics knowledge base and KEPTMs for ethics domain are researched 
 **Iteration 1**: Therefore, to improve the ethical sweeping capacity of Chinese NLP techniques and their effectiveness in the ethical domain, Chinese knowledge bases on ethics and KEPTMs for the ethical domain are sought. 
 **Iteration 2**: Therefore, to improve the ethical brushing capacity of Chinese NLP techniques and their effectiveness in the ethical domain, Chinese knowledge bases on ethics and KEPTMs for the ethical domain are sought. 
 **Iteration 3**: Therefore, to improve the brushing capacity of Chinese NLP methods and their effectiveness in the ethical domain, Chinese knowledge bases on ethics and KEPTMs for the ethical domain are sought. 
 **Iteration 4**: Therefore, to improve the brushing capacity of Chinese NLP methods and their effectiveness in the ethical domain, Chinese knowledge bases on ethics and KEPTMs for the ethical domain are sought. 
 **Iteration 5**: Therefore, to improve the brushing capacity of Chinese NLP methods and their effectiveness in the ethical domain, Chinese knowledge bases on ethics and KEPTMs for the ethical domain are sought. |
| Sample 3: |
| **Original input**: I kindly request a modification to the due date for my assignment. According to the previous thread, I was informed that I would receive an extension of 33 hours for this assignment. However, as of now, my Moodle account still indicates the original due date. 
 **Iteration 1**: Please extend the deadline for my assignment. I was informed that I will receive a 33 hour extension for this assignment, however, it still shows the initial date on my Moodle account. 
 **Iteration 2**: Please extend the deadline for my assignment. I was informed that I will receive a 33 hour extension for this assignment, or this amount is still visible on my Moodle account. 
 **Iteration 3**: Please extend the deadline for my assignment. I was informed that I will receive a 33 hour extension for this assignment, this quantity is always visible on my Moodle account. 
 **Iteration 4**: Would you like to request an extension of the deadline for my mission? I am informed that I will obtain a 33 hour extension for this mission, this quantity is still visible on my Moodle account. 
 **Iteration 5**: Did you want to request an extension of your mission deadline? I was informed that I will obtain a 33 hour extension for this mission, this amount is visible on my Moodle account. |

Table 18: Three original samples and their corresponding $k$-iteration from GPT4.

| Sample 1: |
|---|

**Original input**: Working in groups of 2 or 4 (even numbers may work better later on), suggest a heuristic to evaluate the utility of a given board configuration.
**Iteration 1**: Working in groups of 2 or 4 (even numbers may work better later), propose a heuristic to evaluate the usefulness of a given board configuration.
**Iteration 2**: Working in groups of 2 or 4 (even numbers may work better later), propose a heuristic to evaluate the usefulness of a given board configuration.
**Iteration 3**: Working in groups of 2 or 4 (even numbers may work better later), propose a heuristic to evaluate the usefulness of a given board configuration.
**Iteration 4**: Working in groups of 2 or 4 (even numbers may work better later), propose a heuristic to assess the usefulness of a given board configuration.
**Iteration 5**: Working in groups of 2 or 4 (even numbers may work better later), propose a heuristic to evaluate the usefulness of a given board configuration.

| Sample 2: |
|---|

**Original input**: Therefore, to improve the ethical coping ability of Chinese NLP techniques and their application effectiveness in the field of ethics, the Chinese ethics knowledge base and KEPTMs for ethics domain are researched
**Iteration 1**: Therefore, to improve the ethical management capacity of Chinese NLP techniques and their application effectiveness in the field of ethics, the Chinese ethical knowledge base and KEPTMs for the field of ethics are being studied.
**Iteration 2**: Therefore, to improve the ethical management capacity of Chinese NLP techniques and their application effectiveness in the field of ethics, the Chinese ethical knowledge base and KEPTMs for the field of ethics are under study.
**Iteration 3**: Therefore, to improve the ethical management capacity of Chinese NLP techniques and their application effectiveness in the field of ethics, the Chinese ethical knowledge base and KEPTMs for the field of ethics are currently under study.
**Iteration 4**: Therefore, to improve the ethical management capacity of Chinese NLP techniques and their application efficiency in the field of ethics, the Chinese ethical knowledge base and KEPTMs for the field of ethics are currently under study.
**Iteration 5**: Therefore, to improve the ethical management capacity of Chinese NLP techniques and their application efficiency in the field of ethics, the Chinese ethical knowledge base and KEPTMs for the field of ethics are currently under study.

| Sample 3: |
|---|

**Original input**: I kindly request a modification to the due date for my assignment. According to the previous thread, I was informed that I would receive an extension of 33 hours for this assignment. However, as of now, my Moodle account still indicates the original due date.
**Iteration 1**: I kindly request a modification of the due date for my assignment. According to the previous thread, I was informed that I would receive a 33-hour extension for this assignment. However, at the present time, my Moodle account still shows the initial due date.
**Iteration 2**: I kindly request a change in the due date for my assignment. According to the previous thread, I was informed that I would receive a 33-hour extension for this assignment. However, at present, my Moodle account still displays the initial due date.
**Iteration 3**: I kindly request a change of the due date for my assignment. According to the previous thread, I was informed that I would receive a 33-hour extension for this assignment. However, at present, my Moodle account still displays the initial due date.
**Iteration 4**: I am kindly requesting a change in the due date for my assignment. According to the previous thread, I was informed that I would receive a 33-hour extension for this task. However, at the present time, my Moodle account still displays the initial due date.
**Iteration 5**: I kindly request a change of the due date for my assignment. According to the previous thread, I was informed that I would receive a 33-hour extension for this task. However, at the present time, my Moodle account still displays the initial due date.

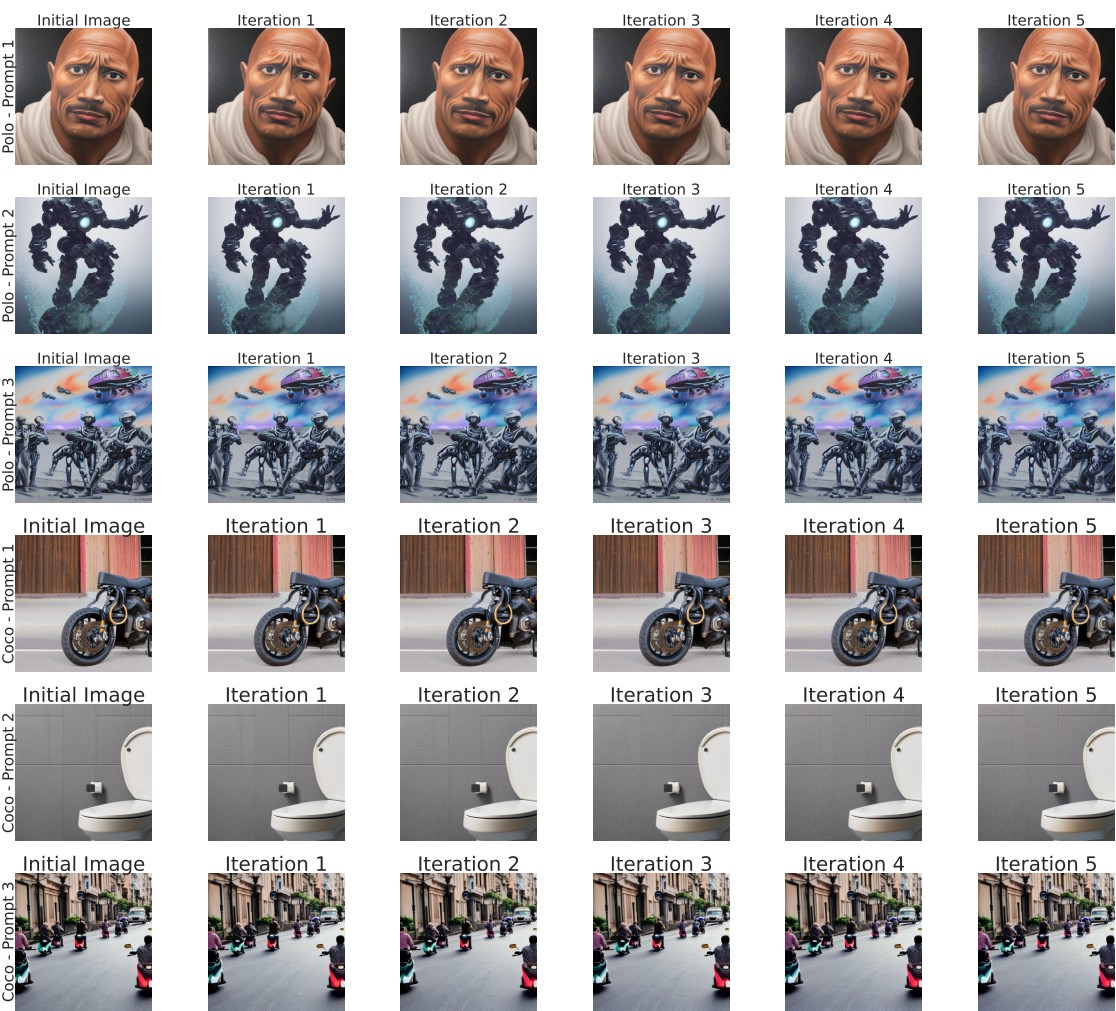

Figure 22: Initial image generation and subsequent re-generations by the SDv2.1 model on Coco and Polo datasets.

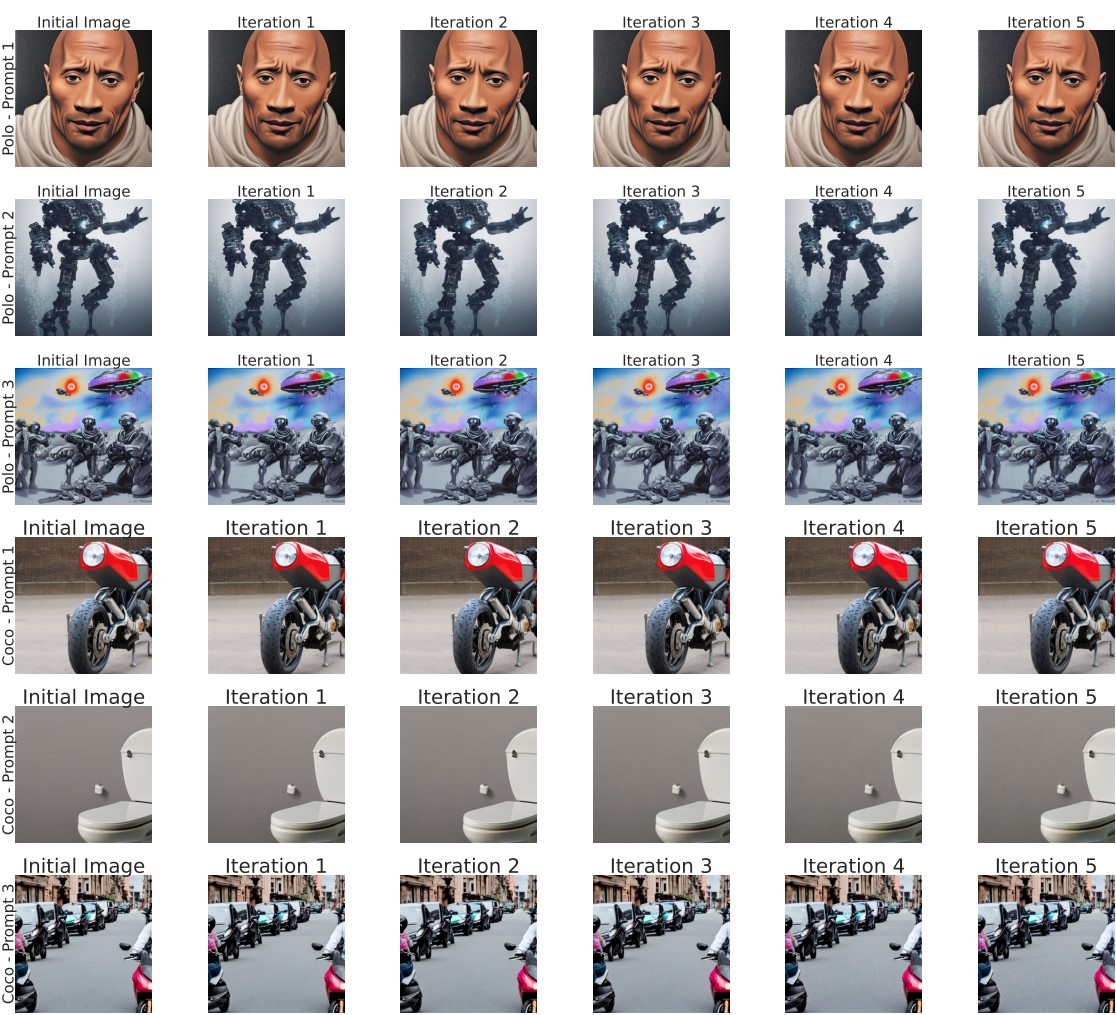

Figure 23: Initial image generation and subsequent re-generations by the SDv2 model on Coco and Polo datasets.

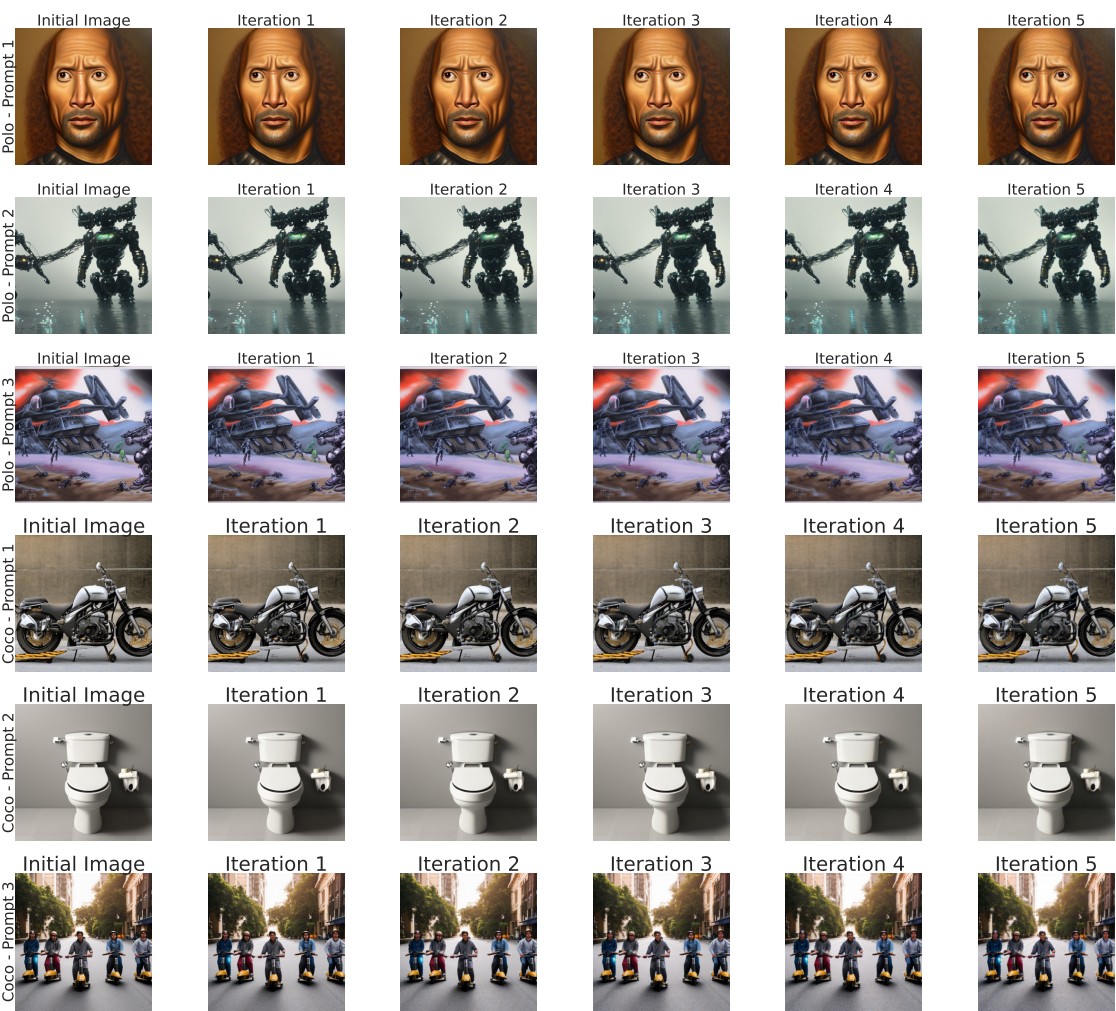

Figure 24: Initial image generation and subsequent re-generations by the SDv2.1B model on Coco and Polo datasets.

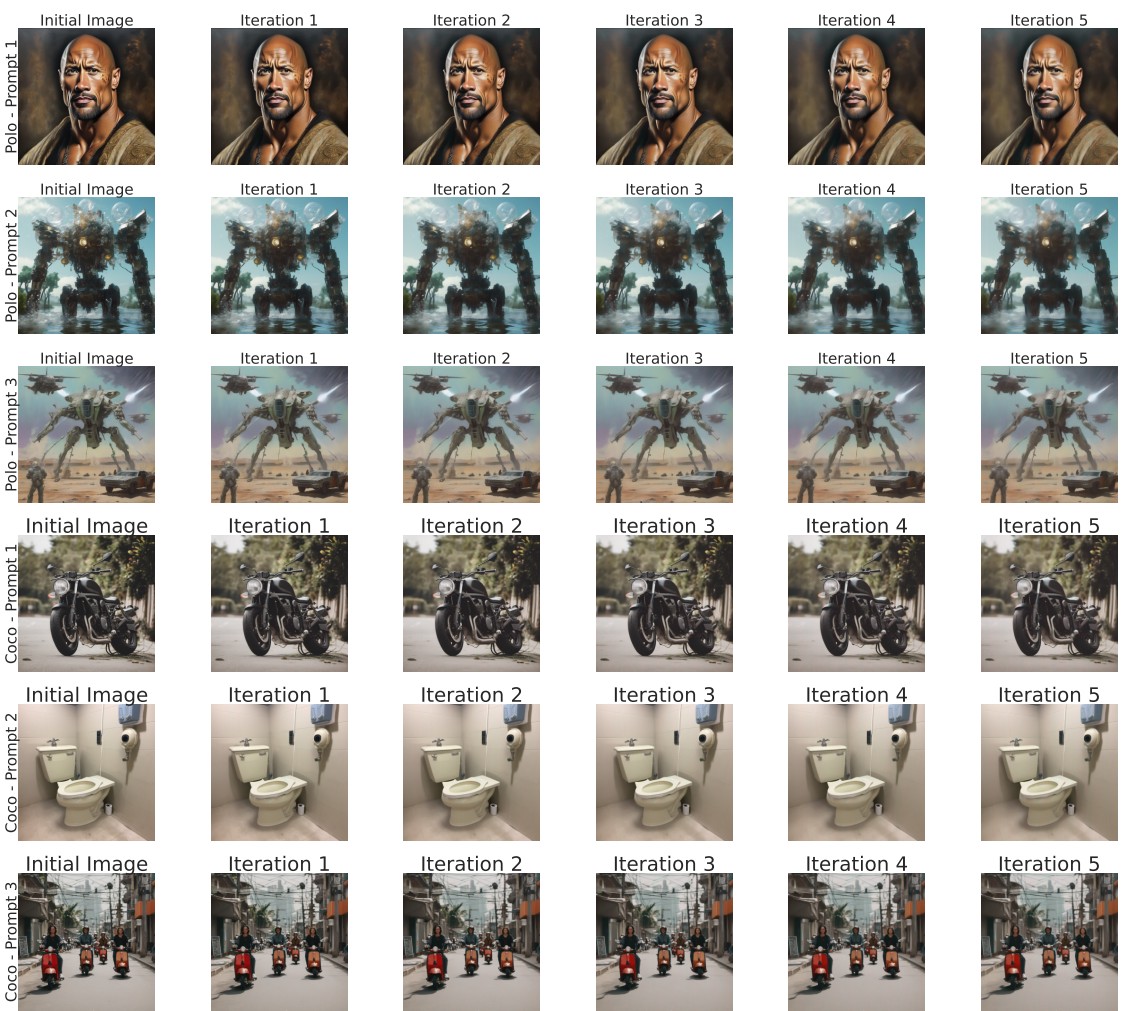

Figure 25: Initial image generation and subsequent re-generations by the SDXL1 model on Coco and Polo datasets.

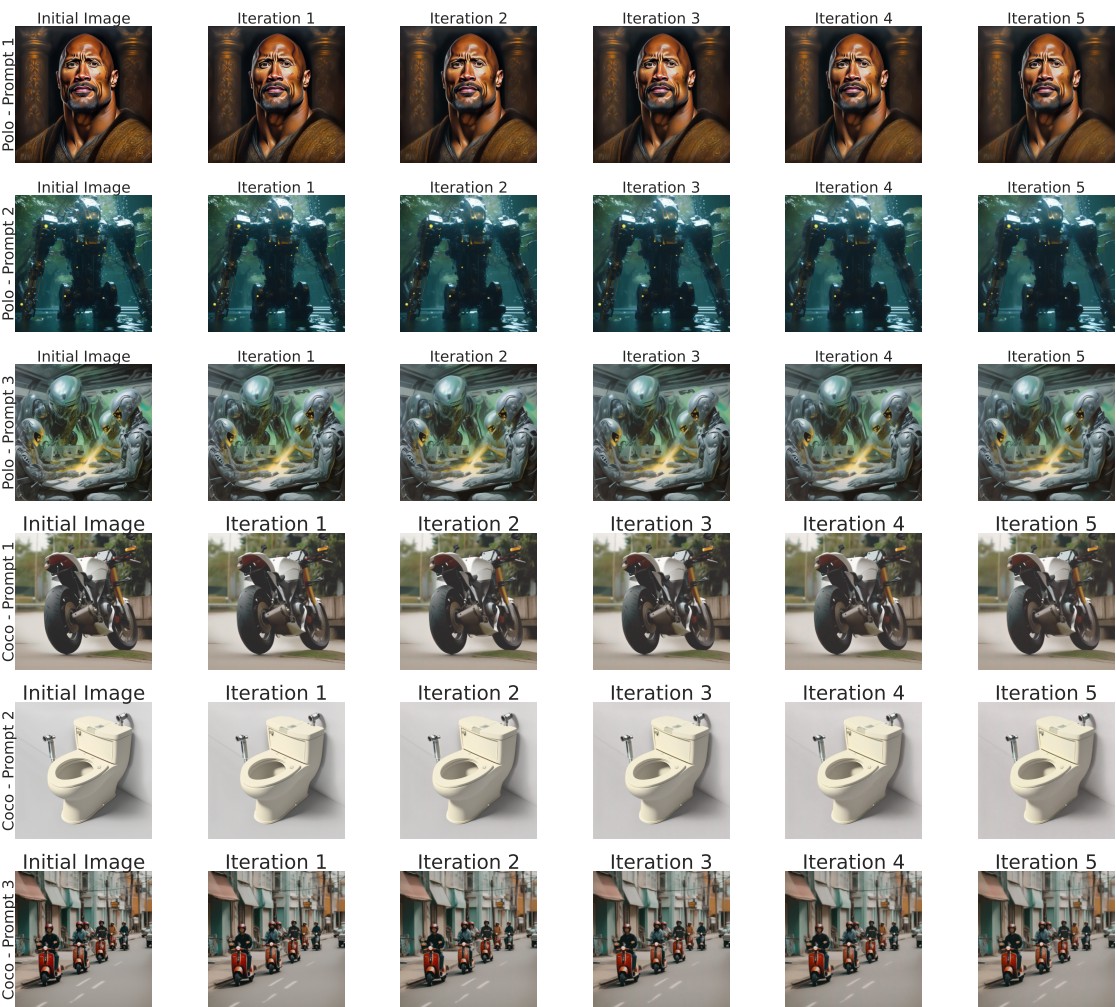

Figure 26: Initial image generation and subsequent re-generations by the SDXL0.9 model on Coco and Polo datasets.

