# OpenReview forum: "Generative Models are Self-Watermarked: Declaring Model Authentication through Re-Generation"
_TMLR — Accepted by TMLR_

### Review · Reviewer_yrvY · 2024-05-23

**Summary Of Contributions:**

The paper presents a technique for identifying the ownership of the output generated from a generative model. The main contribution of the paper is that it identifies a property in the generative model that is common across both text-generating models and image generating models. Given a specific output, this property is used to distinguish between the authentic model and the contrasting model.

Overall I think the paper have some interesting observations, however it needs several changes to be convincing enough research direction.

**Audience:**

Yes

**Claims And Evidence:**

No

**Requested Changes:**

As mentioned above:
1. Changes in theory part
2. Changes in the writing
3. More discussion on comparison to watermarking

**Strengths And Weaknesses:**

I will present some issues and suggest improvements:

Theory:
There are two main issues that I see in the theory part
1) G is used to denote the generative model and the same notation G is used to denote the relation x_{k+1} = G (x_k). The actual implementation of the algorithm uses G with specific settings (e.g. translation, paraphrasing for text with specific prompt template). Thus if G is the actual generative model then for the equation the technique uses a specific parameterization G^*  that can be obtained from G  x_{k+1} = G^* (x_k). Fix this inconsistency across the paper and use  a parameterized notation for this function.

2) The Lipschitz constant theory seems quite redundant at this point since it is not connected well with the technique in the paper. It feels like the paper introduces some math just for sake of writing some math. There is one section in the appendix that shows that the Lipschitz constant under certain setting is less than 1. However, there are no theoretical or empirical justifications overall why the Lipschitz constant is less than 1 for the authentic models. Also, how does this introduced theory+experiments justifies that the edit-distance between iterations does not converge for non-authentic models?

Writing:
The writing can be considerably improved. There are parts of the paper which has too much text and parts that do not have enough text. The algorithms 1 and 2 are not described.  The theory part is not connected to the evaluation and actual work of this paper. The related work is too long, and there is little comparison to the current work. I’ll suggest moving large portion of the related work to appendix or later in the paper and focus more on section 3 and 4. Even the Tables in Section 5 can use a little more description.

Evaluation:
The introduction compares the current work with watermarking, and there is no comparison to the current SOTA watermarking technique. The watermarking techniques such as (Kirchenbauer et. al. 2023) attempt to have a very low false positive rate, in comparison the proposed technique does have a significantly high false positive rate. I appreciate the ablations and additional experiments performed in the evaluation section, but I think the main evaluation still lack the evidences to prove the claims made in the introduction of the paper.

---

> ### Author Response · Authors · 2024-05-28
> **Response to Reviewer yrvY**
>
> Thank you for your review and suggestions.
>
> ---
>
>
> > Theory ($G^*$ notation):
>
> Thank you for this suggestion. We will incorporate the $G^*$ notation in our algorithms.
>
> ---
>
>
> > Theory (Lipschitz Constant):
>
> The convergence of our iterative approach can be explained by the Fixed-Point Theorem, which applies to L-Lipschitz continuous functions. Empirically verifying the condition of the theory within our application should be regarded as strong evidence of the feasibility of the approach. It is important to note that large foundation model service providers hold the authority and responsibility to verify such property in their models to build confidence in the identifiability of their model fingerprints.
>
> ---
>
>
> > Writing:
>
> We will refine our paper to enhance the readability of both the theory and experiments, following your suggestions.
>
> ---
>
>
> > Evaluation and baseline:
>
> We view our work as akin to fingerprint detection, which is quite different from conventional watermarking.
>
> + We would like to emphasize that most existing watermarking approaches require either editing the model parameters or post-editing the model outputs, which often degrade the utility of the models. In contrast, our approach involves applying the model multiple times, ensuring the same level of output quality, as illustrated by the images in Figure 4.
>
> + Our work can operate under a black-box setting, enabling authorship authorization via API calls from third-parties. This approach also allows us trace back the plagrism in `older' models that do not explicitly encode watermarks.
>
> + Furthermore, our approach serves as a framework for generative models across various modalities. Most existing watermarking techniques are designed for a single modality, such as the approach by Kirchenbauer et al. 2023, which is specific to the text.

---

### Review · Reviewer_6KNk · 2024-06-06

**Summary Of Contributions:**

The paper addresses the important challenge of identifying whether a data (single sample) is generated using a specific generative model. Compared to traditional use of watermarking, this paper aims at using "inherent fingerprints" in the outputs without changing the models. The verification is accomplished through regeneration. This avoids changing model and potentially compromising the output, changing the output itself, or use of any additional classification model. Further, the paper uses iterative re-generation to make the model's fingerprint more pronounced. The overall idea is simple: if a model is asked to regenerate its output, it would have a shorter distance between the regenerated output and original output, when compared to a contrastive/different model regenerating the output.

**Audience:**

Yes

**Broader Impact Concerns:**

There are no broader impact concerns. The success of the proposed method would help ensure one can check IP compliance when generative models are used to create artifacts.

**Claims And Evidence:**

Yes

**Requested Changes:**

1. See weakness 1 on the type signature error for G_a. It should be easy to fix in presentation.

2. How does using the model itself in paraphrasing impact the watermarking verification? Note that the rest of analysis in the paper is assuming paraphrasing is independent from the model.

**Strengths And Weaknesses:**

Strengths:

+ The presented approach does not require modifying the model and hence, consequently, it does not compromise the output quality.

+ The approach does not require changing the output by embedding any artificial watermark into it which would again lower the quality and also make it susceptible to forgery attacks.

+ The presented approach is computationally efficient and requires minimal overhead for verification relying on the generative model itself.

+ The use of fixpoint to iteratively magnify the inherent watermark is very interesting.

+ The experimental results are encouraging and the paper considers diverge use cases/scenarios for evaluation.

Weaknesses:

- One limitation in presentation is how the model is used for re-generation. The G_a receives prompt to create output x_a from prompt x_p in algorithm 1 in the initial state but after that the signature of G_a changes to take x_a as input. If x_p is text and x_a is image, this would be a type error and not feasible. Looking at the example case studies in the experiment section, this "paraphrasing" becomes more clear. For the translation task, the paper utilizes "round-trip translation as a way for paraphrasing"

---

> ### Author Response · Authors · 2024-06-17
> **Response to Reviewer 6KNk**
>
> Thank you for your appreciation and valuable feedback.
>
> ---
>
> > Type Signature Difference for using $G_a$ in different steps:
>
> To clarify, our process ensures consistency in output data types, either images or texts, expended on the generative models. On the other hand, our method preserves the generative model's diverse utility by accepting multiple modal prompt inputs in the initial generation rounds.
> We acknowledge your concerns about the potential confusion. We will add the clarification to Section 4.1 and highlight the teaser figure.
>
> Note that round-trip translation was one of the standard ways to 'paraphrase' text in the pre-LLM era. Zero-shot paraphrasing using prompts could serve as another way to paraphrase texts, as studied in Table 4.
>
>
>
> ---
>
> > *"How does using the model itself in paraphrasing impact the watermarking verification?"*
>
> The use of the model itself for paraphrasing positively impacts the watermarking verification. When the model paraphrases its own output, it embeds its unique inherent fingerprints into the regenerated content. This process relies on the model's distinctive characteristics, which remain consistent through iterations. As described in Section 4.3, this iterative re-generation amplifies the model's fingerprints, making them more pronounced and easier to detect.
>
> The methodology treats paraphrasing as a process where the model reconstructs content based on its internal knowledge and style. Whether the paraphrasing is independent or performed by the model itself, the unique fingerprints are preserved. Thus, the verification process remains reliable and effective, as it continues to rely on the consistent characteristics of the model's outputs. This ensures that the watermarking verification is robust and not biased by the paraphrasing process, maintaining the integrity of the authentication mechanism.
>
> ---

---

### Review · Reviewer_GjWk · 2024-06-07

**Summary Of Contributions:**

This paper focuses on the problem of watermarking or identifying if the a given image was generated by a blackbox GenAI model. Specifically, it focuses on image generation, and three text generation tasks: machine translation, paraphrasing, and summarization. The task is formulated as being able to verify whether an image was generated by a candidate GenAI model in contrast to other GenAI models.

The crucial idea that is being empirically tested behind the proposed approach is that one-step regeneration/paraphrasing of the query input will show a slow deviation under the original (authentic) model and large deviation under contrasting GenAI models. Moreover, another presumption that is being tested is that multiple steps of regeneration/paraphrasing under a model will enhance the distinguishing stylistic features of the model enabling greater distinguishability from other models. A result of fixed point convergence assuming Lipschitz continuity of the regernation function is also provided. The experiments are conducted on image generation tasks, and text generation tasks in which following aspects are empirically tested:

a) reduction of distance between subsequent iterations of regeneration as the number of regenerations increases,

b) discrepancy: distance between one-step regeneration under the authentic model and one-step regeneration under a control model,

c) verification: measuring distinguishability of generation via an authentic model vs. a contrast model based on a distance-ratio threshold that is empirically set.

Additional experiments on robustness evaluation are also presented. In this setting, a generation from an authentic model is perturbed either by adding noise of some kind or by paraphrasing via another GenAI model. The task is to verify the authentic model of this perturbed image.

**Audience:**

Yes

**Broader Impact Concerns:**

I could not find a section discussing broader impacts of this approach. Since the approach explicitly focuses on watermarking and copyright/attribution, some discussion around security concerns related to findings in the paper about regeneration would be preferable.

**Claims And Evidence:**

Yes

**Requested Changes:**

Please address concerns listed in the weaknesses (evaluation and claims) section. Especially, address the first three points. I could accept that those experiments might not be in the scope of the current work. If this is the case, then the language must be changed and the claims be adjusted accordingly. Regardless, some discussion around these concerns should be there in the paper.

**Strengths And Weaknesses:**

**Strengths**:

-- The paper is well written and easy to follow.

-- The experiments are well-designed and provide a reasonable idea about the efficacy of the proposed approach.

-- The general trends in the results seem to validate the central hypothesis that one step regeneration under an authentic model differs from one-step regeneration under a contrast model. This effect intensifies as the number of regenerations increase for many tasks and models.

-- To my knowledge, this is a novel approach to test watermarking and the results show interesting behavior of the GenAI models studied in the paper.

**Weaknesses (evaluation and claims)**:

-- The verification evaluation setup can be improved. In the current version, the authentic model is compared against contrast models. But the actual practical question is to identify which  GenAI model was used to generation the artifact in question. One would need to multiple pairwise tests in which all the models are considered to be candidates for the original model and then are compared across contrast models. Finally, the statistics would be aggregated across all these tests to identify one of the models as the authentic model. Accuracy, precision, and recall in this setup would be more informative about the efficacy of the approach to identify the provenance of the artifacts.

-- On a related note, the proposed experimental setup relies on verifying an authentic model from a pre-identified set of GenAI models. This poses a problem as the number of blackbox GenAI models increase in practical scenarios. Moreover, continued access to blackbox models is assumed. But many such blackbox models served by organizations are periodically updated. For example, would this approach work if the GPT_3.5 model is updated in a few weeks from when the artifact was generated?

-- It is unclear how one would differentiate purely human generated artifacts from GenAI artifacts with this approach. One-step regeneration hypothesis, if it holds, can reliably distinguish between the models only when one of them is an authentic model. What happens when neither of the models being compared are authentic?

-- The theoretical result makes a strong assumption about Lipschitz continuity of one-step regeneration functions. This assumption is not convincingly justified either theoretically or empirically.

-- Image generation experiments only focus on models like Stable Diffusion. The regeneration capabilities might differ significantly across image models. Would this generalize to other kinds of image generation GenAI models.

-- BLEU and ROUGE scores are corpus-level scores in general and are. Typically unreliable when used for individual instances. Other distances like Levenshtein edit distances, hamming distances, etc. would be informative for text-based tasks.

-- Contrast model perturbation as is done for the image models, isn’t done for the text-based experiments.

**Weaknesses (results)**:

-- Cohere model is difficult to distinguish. This is linked to my concerns above. As newer models emerge, it is unclear if this approach would sufficiently generalize and scale.

-- Paraphrasing/perturbation seems to decrease the efficacy of the proposed approach.

---

> ### Author Response · Authors · 2024-06-17
> **Response to Reviewer GjWk (Part 1)**
>
> Thank you for your thoughtful and detailed feedback. We appreciate your insights and suggestions.
>
> ---
>
> > Verification Evaluation Setup:
>
> *"One would need to multiple pairwise tests in which all the models are considered to be candidates for the original model and then are compared across contrast models."*
>
> We have provided pairwise tests that compare original models and their contrasts, in Table 2, Table 3 and Table 4. Note that the results from multiple contrast models are essentially the averaged results by all involved contrast models, regarding precision and recall.
>
> *"Accuracy, precision, and recall in this setup would be more informative about the efficacy of the approach"*
>
> We have differentiated the evaluation into precision and recall to reflect the 'accuracy' and 'coverage' performance in verification.
>
> ---
>
> > *"A(a)s the number of blackbox GenAI models increase in practical scenarios," "Would this approach work if the GPT_3.5 model is updated in a few weeks from when the artifact was generated?"*
>
> Thank you for your suggestion. We have progressively followed and explored this scenario. Specifically, the published sentences were generated by GPT-3.5-turbo (0613). During the verification stage, we used two other versions of GPT-3.5: GPT-3.5-turbo (1106) and GPT-3.5-turbo (0125) to verify the authorship claim.
>
> | $G_a$ \ $G_c$             | M2M | mBART |Cohere|
> | :---------------- | :------: | :----: |:----: |
> | GPT3.5-turbo (0613) |  92.0 | 91.0 | 96.0 |
> | GPT3.5-turbo (1106) |  86.0 | 85.0 | 93.0 |
> | GPT3.5-turbo (0125) |  86.0 | 87.0 | 95.0 |
>
> The table above shows that despite a minor performance drop, our approach is effective in differentiating the contrast models using the updated versions.
>
> ---
>
> > Human vs. GenAI Generated Artifacts
>
> *"how one would differentiate purely human generated artifacts from GenAI artifacts with this approach"*
>
> In Section 5.4, we tested the possibility of differentiating human-generated artifacts from AI-generated artifacts. Our findings indicate a significant difference in one-step re-generation distances between these two types of artifacts. Specifically, when AI-generated images are re-generated using our Generation Algorithm 1, the distances remain low. In contrast, re-generating natural (human-generated) images results in significantly higher distances, even with the most advanced AI models. This demonstrates that natural images exhibit different characteristics when processed by our regeneration method compared to AI-generated images.
>
> To further validate our approach, we conducted an experiment to classify them based on various thresholds. We calculated precision, recall, and F1 scores for these classifications. We also translated these thresholds into $\mu \pm x \cdot \sigma$, where $\mu$ and $\sigma$ are mean and variance of corresponding Nature (N) and AI (A) sets. The following table shows that most results are quite decent and the best performance is achieved when the threshold is set to the middle.
>
>
> | Threshold | Precision | Recall | F-1  | x (N) | x (A) |
> |-----------|-----------|--------|----------|-------------|--------|
> | 0.02      | 0.7418    | 1.000  | 0.852   | -1.88     | 0.09 |
> | 0.03      | 0.8701    | 0.991  | 0.927   | -1.58     | 0.81 |
> | 0.04      | 0.9280    | 0.942  | **0.935**   | -1.28     | 1.54 |
> | 0.05      | 0.9572    | 0.850  | 0.900   | -0.98     | 2.26 |
>
>
>
> ---
>
> > Lipschitz Continuity Assumption
>
>
> The convergence of our iterative approach can be explained by the Fixed-Point Theorem, which applies to L-Lipschitz continuous functions. Empirically verifying the condition of the theory within our application should be regarded as strong evidence of the feasibility of the approach. More details of the estimation of L are provided in Appendix B.3.
>
> It is important to note that large foundation model service providers hold the authority and responsibility to verify such property in their models to build confidence in the identifiability of their model fingerprints.
>
> ---
>
> > Generalization to Other Image Models
>
> Our current work verifies the state-of-the-art image generative models (largely based on diffusion architecture) and text generative models (largely based on transformer architecture) under both black-box and white-box settings.
>
> The current framework requires support for 'in-painting' masked pixels. We consider this a standard feature in most image generative model, regardless of their architecture. We will mention this in the limitation section, although we do not anticipate it being a significant drawback.
>
> ---

---

> ### Author Response · Authors · 2024-06-17
> **Response to Reviewer GjWk (Part 2)**
>
> (to continue)
>
> ---
>
> > Text-Based Distance Metrics
>
> *"BLEU and ROUGE scores are corpus-level scores in general and are. Typically unreliable when used for individual instances."*
>
> BLEU and ROUGE are used to evaluate the match of two texts, regardless of they are sentences or paragraphs. They are arguably the predominating evaluation metrics in NLP community for evaluating Machine Translation, Summarisation, and other generative tasks.
>
> *"Other distances like Levenshtein edit distances, hamming distances, etc. would be informative for text-based tasks."*
>
> Our work shows a promising fingerprint verification framework using basic metrics. This framework could potentially be improved with more advanced metrics, offering a feasible direction for future research. Nevertheless, in line with this suggestion, we have employed the edit distance as the distance metric.
>
> | $G_a$ \ $G_c$             | M2M | mBART |Cohere| GPT3.5-turbo|
> | :---------------- | :------: | :----: |:----: |:----: |
> | M2M        |   -   | 89.0|  98.0| 91.0 |
> | mBART           |   90.0 | - | 98.0 | 87.0 |
> | Cohere    |  74.0   |  83.0 | - | 67.0 |
> | GPT3.5-turbo |  92.0 | 91.0 | 96.0 | -|
>
> As shown in the above table, edit distance is also effective in our framework.
>
> ---
>
> > *"Contrast Model Perturbation for Text Experiments"*
>
> We have studied the perturbation for the text generation case and presented the results in Table 5. Hope this can address your concern.
>
> ---
>
> > *Cohere model is difficult to distinguish.*
>
> In Appendix C.3.1 and Table 16, we discussed the difficulty in distinguishing the Cohere model is largely due to its relatively poor translation quality, which leads to relatively slower convergence than other models. However, our approach can significantly improve verification performance after multiple iterations, as the results of k=5 demonstrated in Table 2.
>
> ---
>
> > *"Paraphrasing/perturbation seems to decrease the efficacy of the proposed approach"*
>
> Paraphrasing is always a killer of watermarks and fingerprints [1,2]. In our work, we demonstrate that our approach is stable to a reasonable amount of noise, in Table 5 and 6. For text generation, our approach remains effective even when 30-40% of the inputs are perturbed. As stated in our submission, perturbing more than 30% can significantly degrade the quality of the generated texts, making them considerably less useful for potential attackers. Similarly, for image generation, Table 6 shows that while our method's precision decreases with increasing brightness perturbation, it maintains reasonable performance up to a certain level of perturbation. These results underscore the robustness of our approach to both text and image perturbations.
>
>
> References
>
> [1] Miranda Christ, Sam Gunn, and Or Zamir. Undetectable watermarks for language models. arXiv preprint arXiv:2306.09194, 2023
>
> [2] Zhao, Xuandong, et al. "Generative autoencoders as watermark attackers: Analyses of vulnerabilities and threats." arXiv preprint arXiv:2306.01953 (2023).
>
> ---
>
> Thank you for your detailed feedback and suggestions. We hope our answers have solved your concerns and we are more than happy to answer follow-up questions.

---

> ### Comment · Reviewer_GjWk · 2024-06-24
> **Thanks for your reply**
>
> Thanks a lot for your detailed reply! I just have one small comment -- Tables 2, 3, and 4 do not necessarily provide an analysis of what I am looking for. The setup I am imagining is: given a query image, make N rows for this image (where N is possible candidates that could have generated this image), then perform comparison against all the contrast models (say N models). Finally based on all the numbers in this N by N matrix, make a decision about where this image came from. Do this for all the images in the dataset. Compute precision, recall, accuracy on this metric. Table 2, 3, and 4 do provide statistics that are related to the process I described above but they don't reflect this process exactly.
> This is an expensive procedure, but one that comes close to addressing realworld scenario settings.

---

> ### Author Response · Authors · 2024-06-27
> **Evaluation via $N\times N$ contrast score matrix**
>
> > "Given a query image, make N rows for this image (where N is possible candidates that could have generated this image) .... Compute precision, recall, accuracy on this metric"
>
> Thanks for the suggestion. We experimented to evaluate the verification performance using all images/sentences generated by multiple generative models, as you suggested. Specifically, for each image/sentence, we obtained N distance scores by querying N models with our one-step re-generation method. We then normalized these distance scores by dividing each by the score from a candidate model, resulting in N contrast scores forming a row in the evaluation score matrix. Repeating this process on all possible candidate models, we created an $N\times N$ matrix. For each row, we calculated the average contrast score across all columns. The model with the largest average contrast score was then considered the predicted authentic model. We present precision, recall, and accuracy for all studied tasks using $K=5$ below:
>
> | __Tasks__             | __Accuracy__| __Precision__ | __Recall__ |
> | :---------------- | :------: | :----: |:----: |
> | Machine Translation |  82.5 |84.6 | 82.5 |
> | Image Generation | 83.1  | 84.6 | 83.1 |
> | Paraphrasing |  69.0 | 69.6 | 69.0 |
> | Summarization |  71.3 | 76.3 | 71.3 |
>
> As shown in the above table, our approach can effectively verify the authentic model among multiple possible candidates. Hope these results can address your concern.

---

### Decision · Action_Editor_MaJv · 2024-07-30

**Recommendation:** Accept with minor revision

**Comment:**

This paper addresses the challenge of identifying whether an image or text was generated by a specific GenAI model, without altering the model or output. The approach hinges on the concept that regenerating or paraphrasing an input through the original model will result in minimal deviation, whereas using a different model will cause significant deviation. The paper empirically tests that iterative regeneration enhances the model’s distinctive features, facilitating greater differentiation. By leveraging the inherent fingerprints in outputs and avoiding additional classification models, the proposed method effectively verifies the origin of the data. The key contribution is a universal property applicable across text and image generation models, allowing for the distinction between the authentic model and others based on the regeneration behavior.

Using generation for watermarking is indeed an interesting idea. However, some issues exist that need to be addressed before publication.

- One of the concerns raised by a reviewer is that the Lipschitz constant is only practically verified. I understand that measuring the Lipschitz constant for a typical Generative AI is not possible (in particular for closed models). In this sense, the claim of theorem 2 is too overclaimed and should be fixed. One suggestion would be to make an assumption that the Lipschits constant is L \in (0,1). Then, add more discussion in 4.3 that this assumption is practically satisfied in many Generative AI models.
- As the authors responded, "We will refine our paper to enhance the readability of both the theory and experiments, following your suggestions." Please revise the paper based on the reviewer's suggestions and improve the readability.
- As pointed out by reviewers, the approach has some limitations. So, please summarize the weakness of the proposed method in either the main text or the supplementary materials.

**Audience:**

Watermarking of generative AI is a hot topic in AI/ML. Thus, this may attract many researchers.

**Claims And Evidence:**

Mostly Yes. The paper conducted many experiments and this supports the idea.

---

> ### Author Response · Authors · 2024-08-29
>
> Responses and actions taken to the suggestions:
>
> | **Reviewer/Editor**      | **Comment/Concern**                                                                                      | **Action Taken**                                                                                           | **Location in Paper**           |
> |-------------------|----------------------------------------------------------------------------------------------------------|-------------------------------------------------------------------------------------------------------------|---------------------------------|
> | **Editor MaJv and Reviewer GjWk**   | The generalizability of the methodology to other image models.                                                     | We have added a discussion in limitation section regarding the methodology's generalizability to other image models.                      | Limitation section                 |
> | **Reviewer GjWk** | Improving the verification evaluation setup and testing the approach's generalizability.             | We have implemented pairwise tests, explored the impact of updates to GPT models, and included a discussion on human vs. AI-generated artifacts. | Section 5.4     |
> | **Reviewer 6KNk** | Highlighted a type signature difference for \( G_a \) in different steps and its impact on watermarking verification. | Clarified the type signature differences in Algorithm 1 and discussed how paraphrasing by the model itself impacts watermarking verification. | Algorithm 1, Section 4.1        |
> | **Reviewer yrvY** | Notation inconsistencies in the use of \( G \) and requested improvements in the theoretical discussion of the Lipschitz constant. | Fixed notation inconsistencies and expanded the discussion on the Lipschitz constant assumption and its relevance to the approach. | Section 4.1 and Section 4.3     |
> | **Reviewer yrvY** | Writing issues including excessive content in the related work section, lack of descriptions for algorithms, and insufficient clarity in table descriptions. | We have added more detailed related work to the appendix, added detailed descriptions of Algorithms 1 and 2, and improved the clarity of table descriptions in Section 5. | Appendix, Algorithms 1 and 2, Section 5 |